# Dynamic Subgroup Identification in Covariate-adjusted Response-adaptive Randomization Experiments

**Yanping Li**
School of Statistics and Data Science
Nankai University
`yanpingli@mail.nankai.edu.cn`

**Jingshen Wang**
Division of Biostatistics
University of California, Berkeley
`jingshenwang@berkeley.edu`

**Waverly Wei** *
Department of Data Sciences and Operations
University of Southern California
`waverly@marshall.usc.edu`

## Abstract

Identifying subgroups with differential responses to treatment is pivotal in randomized clinical trials, as tailoring treatments to specific subgroups can advance personalized medicine. Upon trial completion, identifying best-performing subgroups–those with the most beneficial treatment effects–is crucial for optimizing resource allocation or mitigating adverse treatment effects. However, traditional clinical trials are not customized for the goal of identifying best-performing subgroups because they typically pre-define subgroups at the beginning of the trial and adhere to a fixed subgroup treatment allocation rule, leading to inefficient use of experimental efforts. While some adaptive experimental strategies exist for the identification of the single best subgroup, they commonly do not enable the identification of the best set of subgroups. To address these challenges, we propose a dynamic subgroup identification covariate-adjusted response-adaptive randomization (CARA) design strategy with the following key features: (i) Our approach is an adaptive experimental strategy that allows the dynamic identification of the best subgroups and the revision of treatment allocation towards the goal of correctly identifying the best subgroups based on collected experimental data. (ii) Our design handles ties between subgroups effectively, merging those with similar treatment effects to maximize experimental efficiency. In the theoretical investigations, we demonstrate that our design has a higher probability of correctly identifying the best set of subgroups compared to conventional designs. Additionally, we prove the statistical validity of our estimator for the best subgroup treatment effect, demonstrating its asymptotic normality and semiparametric efficiency. Finally, we validate our design using synthetic data from a clinical trial on cirrhosis.

## 1 Introduction

Most clinical trial designs adopt "one-size-fits-all" rules for treatment assignment and evaluation based on models that ignore patient heterogeneity. This approach is disconnected from medical practice in recent years, where physicians use each patient's diagnosis and prognostic variables to

---

*Corresponding author

38th Conference on Neural Information Processing Systems (NeurIPS 2024).

make personalized, precision medicine treatment decisions. As such, identifying patient subgroups with differential responses to a treatment plays a pivotal role in designing randomized clinical trials [28, 18, 2, 39]. Adaptive clinical trials–that allow randomization probabilities to be adaptively optimized during the trial based on sequentially accrued data–have received much attention due to their potential advantages in promoting precision health. Nevertheless, these trials often involve pre-specifying the patient subgroups to be analyzed [3, 38]. This approach not only leads to inefficient use of experimental efforts but may also reduce the statistical power to detect non-pre-specified subgroups that exhibit high effect sizes. Consequently, there is a pressing need for novel and statistical clinical trial designs for dynamic subgroup identification to address these issues.

In this paper, we propose a novel statistical design that dynamically performs subgroup identification in CARA experiments. Our contributions are summarized as follows:

1. From the design perspective, there are three highlights of our design: (i) Our design facilitates dynamic identification and sequential refinement of the best subgroups within the framework of covariate-adjusted response-adaptive (CARA) experiments (Section 4). This adaptive setting enhances the ability to identify and adjust to the best-performing subgroups over time. (ii) Unlike traditional designs that focus on identifying a single best-performing subgroup, our design is tailored to identify the best set of subgroups with competitive performance (Section 3). This broader objective makes our approach suitable for more general application settings. (iii) Our design demonstrates a higher probability of correctly identifying the best subgroups compared to conventional designs. This efficiency ensures that experimental efforts are utilized more effectively.

2. From a theoretical perspective, our proposed design strategy accommodates tied treatment effects among candidate subgroups, whereas many existing methods demand that these effects be distinctly separated. Additionally, our algorithm involves resampling in the presence of dependent data structures caused by adaptive treatment allocation, presenting technical challenges in proving statistical validity. We overcome these challenges and show that the best subgroup identified by our design asymptotically converges to the true set of best subgroups (Theorem 1). We also demonstrate that our proposed design strategy converges to the oracle design, which is the optimal design under the setting that the underlying data-generating distribution is known (Theorem 2). Furthermore, we establish valid statistical inference for the treatment effect of the identified best subgroup and demonstrate that our constructed estimator is semiparametrically efficient (Theorem 3).

In comparison to the existing literature, our design strategy is closely related to CARA designs. Originating from response-adaptive randomization (RAR) designs [5, 33, 46]. Heuristically, CARA incorporates covariate information along with treatment assignment probabilities based on observed outcomes [14, 32, 30, 13]. Building upon RAR designs, CARA designs utilize both outcome and covariate information to optimize for design objectives. [15] introduces a family of CARA designs that balance efficiency and ethics objectives. Further generalizations to incorporate semiparametric estimates have been explored by [49]. Related developments in CARA designs include [22, 40, 48, 47]. However, conventional CARA designs often consider subgroups to be pre-specified, which is different from our goal of data-adaptively identifying the best subgroups.

Our proposed method also connects with the literature on subgroup identification. In post-hoc analyses using previously collected data, the first line of work uses data from prior randomized controlled trials. [20] employs clustering techniques based on randomized controlled trial data. A comprehensive review can be found in [23]. The second line of work uses data from prior observational studies. [43] identifies subgroups from existing observational data using machine learning. [36] introduces a causal inference tree approach for subgroup identification, which requires specifying the conditional distribution of the outcome given covariates. Similarly, [45] develops causal inference tree types of algorithm that include double robust estimators for constructing subgroup-specific splitting criteria. Other tree-based approaches for subgroup identification include [17, 24, 16]. Besides the tree-based approaches, [10] develops a subgroup identification method within the value function framework. While post-hoc subgroup analyses do not require new data collection, they often rely on untestable causal assumptions, limiting the credibility of causal conclusions. For instance, the unconfoundedness assumption necessary for causal inference in observational studies assumes random treatment assignment based on observed confounders, but unmeasured confounders can

compromise these conclusions. Conversely, in randomized experiments, valid causal conclusions do not depend on such assumptions.

In the adaptive experiment literature, relatively few methods have been developed for identifying the best subgroups. While [11] is in the adaptive experiment setting and proposes a Bayesian adaptive design that sequentially revises treatment allocation to identify the effective subgroup-treatment pairs, their approach is carried out under the Bayesian framework, relying on the specification of prior distributions and does not provide theoretical justification for the identified subgroups. In contrast, our design is aligned with the frequentist framework and is model-free, avoiding any parametric modeling assumption on the joint distribution of potential outcomes and covariates, and providing theoretical investigations from three aspects.

As our method aims to identify the best subgroups, it shares some similarities with the multi-armed bandit (MAB) literature. Notable MAB algorithms, such as the Thompson sampling method [35], the $\epsilon$-greedy algorithm, and the upper confidence bound algorithm [37], focus on identifying the best arm. Similar to the contextual bandit literature [21, 6, 1], our design also incorporates covariate information. However, our design objective diverges significantly from those in traditional MAB approaches as we seek to identify the best subgroups rather than the best arm. Moreover, recognizing that randomized experiments can be time-consuming, costly, and may result in adverse outcomes for patients if treatments are ineffective, our approach seeks to efficiently allocate experimental resources within a constrained budget to identify the most beneficial subgroups. Specifically, we focus on scenarios in which the cost per experimental unit (e.g., per patient) is significant. For instance, in clinical settings, randomized experiments are often expensive due to the substantial costs of treatment medications. As a result, the primary resource constraint in our framework is the limited number of treatments that can be administered, underscoring the need for a resource-efficient experimental design.

## 2 Formulation of CARA

In this section, we shall introduce the formulation of our covariate-adjusted response-adaptive randomization (CARA) experiment framework.

We enroll participants sequentially across $T$ stages, where $T < \infty$. Denote the total number of enrolled participants as $N = \sum_{t=1}^{T} n_t$, where $n_t$ is the number of participants in Stage $t$, for $t = 1, \ldots, T$. The cumulative sample size up to Stage $t$ is denoted as $N_t = \sum_{s=1}^{t} n_s$. In Stage $t$, we denote the treatment assignment status of participant $i$ as $D_{it} \in \{0, 1\}$, $i = 1, \ldots, n_t$, where $D_{it} = 1$ denotes the treatment arm, and $D_{it} = 0$ denotes the control arm. The observed outcome is denoted as $Y_{it} \in \mathbb{R}$. We follow the Neyman-Rubin causal model [27, 34] to define $Y_{it}(d)$ as the potential outcome we would have observed if participant $i$ receives treatment $d$ at Stage $t$, for $d \in \{0, 1\}$. The observed outcome can then be represented as

$$Y_{it} = D_{it} Y_{it}(1) + (1 - D_{it}) Y_{it}(0), \quad i = 1, \ldots, n_t, \quad t = 1, \ldots, T.$$

We assume that the outcomes are observed without delay, and their underlying distributions do not shift over time [14].

In CARA experiments, covariate information is also available to practitioners. We denote the covariate information for participant $i$ as $X_{it} \in \mathbb{R}$ and assume the covariate space $\mathcal{X}$ can be partitioned into $m$ regions, denoted as $\{\mathcal{S}_j\}_{j=1}^{m}$. In clinical settings, each partition of the sample space is commonly referred to as a subgroup [3, 19, 44]. We denote the number of subjects enrolled in subgroup $j$ at Stage $t$ as $n_{tj} = \sum_{i=1}^{n_t} \mathbb{1}_{(X_{it} \in \mathcal{S}_j)}$ and the cumulative sample size for group $j$ up to Stage $t$ is $N_{tj} = \sum_{s=1}^{t} n_{sj}$. Denote the total number of subjects enrolled in subgroup $j$ as $N_j = N_{Tj}$.

As we are interested in assessing the effectiveness of the treatment in each subgroup, we define the subgroup average treatment effect as

$$\tau_j = \mathbb{E}[Y_{it}(1) - Y_{it}(0) | X_{it} \in \mathcal{S}_j], \quad j = 1, \ldots, m.$$

Due to the adaptive nature of CARA experiments, practitioners can sequentially revise treatment allocation based on outcome and covariate information accumulated during the experiment. Formally, we define the treatment assignment probability for participants in subgroup $j$ as

$$e_{tj} = \mathbb{P}(D_{it} = 1 | X_{it} \in \mathcal{S}_j, \mathcal{H}_{t-1}), \quad t = 1, \ldots, T, \quad j = 1, \ldots, m,$$

where $\mathcal{H}_{t-1} = \{(Y_{is}, D_{is}, X_{is})_{i=1}^{n_s}\}_{s=1}^{t-1}$ denotes the historical information up to Stage $t-1$. In CARA experiments, we aim to dynamically revise $e_{tj}$ to reach desired design goals, which shall be introduced in the following section.

# 3 Design objective for best subgroups identification

In real-world applications, suppose we start with $m$ subgroups, and practitioners may only aim to find subgroups with the largest treatment effects. It is possible that the best-performing subgroup is not unique. For example, the FORTE trial is a clinical trial aiming to investigate the treatment effect of carfilzomib-based induction–intensification–consolidation regimens on a patient's progression-free survival rate [25]. Instead of reporting the single best subgroup with the largest survival rate improvement, the trial reports that both the risk myeloma patient subgroup and high-risk patient subgroup show similar survival rate improvement.

Without loss of generality, suppose the population subgroup treatment effects follow the order $\tau_1 \geq \tau_2 \geq \ldots \geq \tau_k > \ldots > \tau_m$. In this case, there are $k$ subgroups exhibiting the largest treatment effects. Thus, it is natural to identify all of the $k$ best subgroups instead of the single best subgroup.

To describe the set of best-performing subgroups, we introduce the concept of a tie set. We denote the tie set of $\tau_1$ as $\mathcal{T}_1 = \{k : |\tau_1 - \tau_k| = o(N^{-1/2}),\ k = 1, \ldots, m\}$ which contains the indices of the tied subgroups. This tie set is also known as the "near tie set" as it captures the subgroups of which the treatment effects lie in the $\sqrt{N}$-local neighborhood of $\tau_1$. We then denote the subgroups that belong to the best set as $\mathcal{S}_{\mathcal{T}_1}$. Note that when $\mathcal{T}_1 = \{1\}$, $\mathcal{S}_{\mathcal{T}_1}$ is equivalent to $\mathcal{S}_1$. The population treatment effect under the best subgroups is defined as $\tau_{\mathcal{T}_1} = \mathbb{E}[Y_{it}(1) - Y_{it}(0)|X_{it} \in \cup_{j \in \mathcal{T}_1} \mathcal{S}_j]$. We further assume that $\tau_{\mathcal{T}_1} > \tau_j$, for $j \notin \mathcal{T}_1$.

Our design objective is to correctly identify all the best subgroups. Mathematically, we aim to maximize the correct identification probability:

$$\max_{e} \mathbb{P}\big(\widehat{\tau}_{\mathcal{T}_1} \geq \max_{j \notin \mathcal{T}_1} \widehat{\tau}_j\big), \text{ where } \mathcal{T}_1 = \{1, \ldots, k\}.$$

Leveraging the large deviation theory [8, 12], we can formulate our design objective as

$$\max_{e} \left\{ \min_{j \notin \mathcal{T}_1} G(\mathcal{S}_{\mathcal{T}_1}, \mathcal{S}_j; e_1, e_j) := \frac{(\tau_j - \tau_{\mathcal{T}_1})^2}{2\big(\mathbb{V}_{\mathcal{T}_1}(e_1) + \mathbb{V}_j(e_j)\big)} \right\}, \quad \leftarrow \text{ Maximize correct selection probability}$$

$$\text{s.t. } \delta \leq e_j \leq 1 - \delta, \qquad\qquad\qquad\qquad\qquad \leftarrow \text{ Feasibility constraints}$$

where $\mathcal{T}_1 = \{1, \ldots, k\}$, $1 \leq k < m$, $\delta \in (0, 1/2)$, and $\mathbb{V}_{\mathcal{T}_1}$ denotes the variance of the best subgroups. Detailed derivations of the equivalence between the correct identification probability and the optimization objective see Appendix (Section C).

However, in practice, solving this optimization problem is challenging. On the one hand, as we do not have knowledge regarding the membership of subgroups that have the largest treatment effects, we do not have any information regarding $\mathcal{S}_{\mathcal{T}_1}$. On the other hand, because experimenters have no prior information about the joint distribution of the subgroup treatment effects, $\tau_j$'s and $\mathbb{V}_j$'s are also unknown. To address these two practical challenges, we propose a dynamic subgroup identification algorithm that can adaptively identify and merge the set of best subgroups. Additionally, the dynamic subgroup identification method operates seamlessly under a CARA experimental strategy, which allows experimenters to sequentially learn the unknown parameters and adjust the subgroup treatment allocation to attain our design objective.

# 4 Proposed design: Dynamic subgroup identification with CARA

In this section, we shall illustrate our proposed dynamic subgroup identification strategy with CARA design in Algorithm 1: the design strategy in Section 4.1, encompassing two sub-algorithms (Algorithm 2 and 3), followed by the statistical inference procedure in Section 4.2. We defer several variations of our algorithm to Appendix (Section J). To clarify our design strategy, we also provide a notation table in the Appendix (Section A, Table 2).

### 4.1 Proposed design strategy

In Stage 1 (line 1-4), we obtain initial estimates of the group-level treatment effect $\widehat{\tau}_{1j}$ and the associated variances $\widehat{\mathbb{V}}_{1j}$. Then, in Stage $t = 2, \ldots, T-1$ (Algorithm 1 line 6–11), we perform three tasks: (1) adaptively update treatment allocation and treatment effect estimates, (2) dynamically identify best subgroups, and (3) select hyperparameters that help with dynamic subgroup identification.

For the first task, we obtain the optimal treatment allocation $\widehat{e}_t^* = (\widehat{e}_{t1}^*, \ldots, \widehat{e}_{tm_t^*}^*)$ by solving the following optimization problem based on sequentially collected data:

$$\max_{e} \ \min_{2 \leq j \leq m_t^*} \frac{(\widehat{\tau}_{t-1,(j)} - \widehat{\tau}_{t-1,(1)})^2}{2\big(\widehat{\mathbb{V}}_{t-1,(1)}(e_1) + \widehat{\mathbb{V}}_{t-1,(j)}(e_j)\big)}, \text{s.t.} \sum_{l=1}^{m_t^*} \widehat{p}_{tl} e_l \leq c_1, \ c_2 \leq e_l \leq 1 - c_2, \ l = 1, \ldots, m_t^*,$$

where $c_1 \in (0,1)$ and $c_2 \in (0, 1/2)$, $\widehat{p}_{tl} = \frac{\sum_{s=1}^{t} \sum_{i=1}^{n_s} \mathbb{1}_{(X_{is} \in \mathcal{S}_{(l)})}}{\sum_{s=1}^{t} n_s}$ is the estimated subgroup proportion. The total number of subgroups after merging the identified tie set in Stage $t-1$ is denoted as $m_t^* = m - |\widehat{\mathcal{T}}_{t-1,1}| + 1$ and the subscript $(j)$ indexes the subgroup with the $j$-th largest estimated treatment effect. The procedure of finding $\widehat{\mathcal{T}}_{t-1,1}$ shall be illustrated in Algorithm 2. In the set of constraints, the first one is the resource constraint, and the second one is the feasibility constraint. Because of the nonlinear objective function of the optimization problem above, we instead work with its equivalent epigraph representation:

$$\widehat{e}_t^* = \arg\max_{e} \left\{ z : \sum_{l=1}^{m_t^*} \widehat{p}_{tl} e_l \leq c_1, \ c_2 \leq e_l \leq 1 - c_2, \ \min_{2 \leq j \leq m_t^*} \frac{(\widehat{\tau}_{t-1,(j)} - \widehat{\tau}_{t-1,(1)})^2}{2\big(\widehat{\mathbb{V}}_{t-1,(1)}(e_1) + \widehat{\mathbb{V}}_{t-1,(j)}(e_j)\big)} - z \geq 0 \right\}. \tag{1}$$

To calibrate for the complete randomized treatment allocation in Stage 1, we require an additional calibration step in Stage $t$:

$$\widetilde{e}_{t,(j)} = \frac{\big(\widehat{e}_{t,(j)}^* N_{t,(j)}\big) - N_{t-1,(j)}(1)}{n_{t,(j)}}, \ j = 1, \ldots, m_t^*, \tag{2}$$

where $n_{t,(j)} = \sum_{i=1}^{n_t} \mathbb{1}_{(X_{it} \in \mathcal{S}_{(j)})}$, $N_{t-1,(j)}(1) = \sum_{s=1}^{t-1} \sum_{i=1}^{n_s} \mathbb{1}_{(X_{is} \in \mathcal{S}_{(j)})} D_{is}$, and $N_{t,(j)} = \sum_{s=1}^{t} n_{s,(j)}$. We then allocate treatments with calibrated probability $\widetilde{e}_{t,(j)}$ and update the subgroup treatment effects as in Eq (8).

**Dynamic identification of the best subgroups (Algorithm 2).** The dynamic subgroup identification algorithm for identifying $\widehat{\mathcal{T}}_{t,1}$ at Stage $t$ involves a resampling step that generates bootstrap samples $\widehat{\tau}_t^\circ$ from a Gaussian distribution centering around $\widehat{\tau}_t$ at Stage 1 and a resampling step that generates bootstrap samples with accrued data $\{\mathcal{H}_s\}_{s=1}^{t}$ at later stages. In line 9, we identify the best subgroups at Stage $t$ as

$$\widehat{\mathcal{T}}_{t,1} = \{k : w_{k,(1)}^\circ = 1, k = 1, \ldots, m_t^*\}, \tag{3}$$
$$\text{where } w_{k,(1)}^\circ = \mathbb{1}\{-c_{\text{L}}^t \cdot N_t^{-\delta} \cdot \widehat{\mathbb{V}}_{t,(1)}^\delta \leq (\widehat{\tau}_{tk}^\circ - \widehat{\tau}_{t,(1)}^\circ) \leq c_{\text{R}}^t \cdot N_t^{-\delta} \cdot \widehat{\mathbb{V}}_{t,(1)}^\delta\},$$

where the distance between the upper and lower bounds of the interval for $w_{k,(1)}^\circ$ is of order $n^\delta$ with $\delta = 0.25$ to guarantee the statistical validity of our proposed procedure and balance the trade-off between bias and variance. Note that the dynamic subgroup identification procedure relies on a pair of hyperparameters $(c_{\text{L}}^t, c_{\text{R}}^t)$, which are selected data-adaptively. In what follows, we shall illustrate the algorithm for selecting these hyperparameters.

**Hyperparameter selection (Algorithm 3).** In line 7 of Algorithm 3, we adopt a bootstrap method and propose several alternative bootstrap methods in Algorithm 5 (line 3) in Appendix (Section J). Algorithm 3 involves a resampling step that generates bootstrap samples $\widehat{\tau}_t^*$ from a Gaussian distribution centering around $\tau_t^*$ at Stage 1. In line 2, we compute $\tau_t^* = (\tau_{t1}^*, \ldots, \tau_{t,m_t^*}^*)'$ as

$$\tau_{tj}^* = \Delta \cdot \frac{\sum_{j=1}^{m_t^*} \widehat{\tau}_{tj}}{m_t^*} + (1 - \Delta) \cdot \widehat{\tau}_{tj}, \ j = 1, \ldots, m_t^*, \tag{4}$$

where $\Delta = \min\{0.99, \frac{\sum_{j=1}^{m_t^*} \widehat{\mathbb{V}}_{tj}}{N_t \sum_{j=1}^{m_t^*} (\widehat{\tau}_{tj} - \widehat{\overline{\tau}}_t)^2} \times N_t^\gamma\}$ and $\gamma \in (0, 0.2)$. We impose a lower bound on $\Delta$ to ensure that $\Delta$ does not equal to 1. We choose $\gamma = 0.05$ in our simulation studies, and our procedure is shown to be not sensitive to the choice of $\gamma < 1$. In line 7, we compute $\widehat{\boldsymbol{\tau}}_t^* = (\widehat{\tau}_{t1}^*, \ldots, \widehat{\tau}_{t,m_t^*}^*)'$ at Stage $t$ for $t > 1$ as

$$\widehat{\tau}_{tj}^* = \Delta \cdot \frac{\sum_{j=1}^{m_t^*} \widehat{\tau}_{tj}^\circ}{m_t^*} + (1 - \Delta) \cdot \widehat{\tau}_{tj}^\circ, \tag{5}$$

where $\widehat{\tau}_{tj}^\circ$ is computed with the bootstrap samples as in Eq (8). In line 10, we compute

$$\widetilde{\tau}_{t,(1)}^* = \sum_{k=1}^{m_t^*} w_{k,(1)}^* \widehat{\tau}_{tk}^* \Big/ \sum_{k=1}^{m_t^*} w_{k,(1)}^*, \quad \mathcal{B}_{tb}(c_{\mathrm{L}}, c_{\mathrm{R}}) = \mathbb{1}(\widetilde{\tau}_{t,(1)}^{*,b} \leq \tau_{t,(1)}^*), \tag{6}$$

where $w_{k,(1)}^* = \mathbb{1}\{-c_{\mathrm{L}}^t \cdot N_t^{-\delta} \cdot \widehat{\mathbb{V}}_{t,(1)}^\delta \leq (\widehat{\tau}_{tk}^* - \widehat{\tau}_{t,(1)}^*) \leq c_{\mathrm{R}}^t \cdot N_t^{-\delta} \cdot \widehat{\mathbb{V}}_{t,(1)}^\delta\}$ and the subscript $(1)$ indexes the subgroup with the largest estimated treatment effect. We also propose a double bootstrap-based alternative hyperparameter selection procedure in Algorithm 4 in Appendix (Section J). Finding the optimal hyperparameters involves minimizing a loss function $L_t(c_{\mathrm{L}}, c_{\mathrm{R}})$ at each Stage $t$, which is defined as

$$L_t(c_{\mathrm{L}}, c_{\mathrm{R}}) = \frac{1}{2}(L_{t0}(c_{\mathrm{L}}, c_{\mathrm{R}}) + L_{t1}(c_{\mathrm{L}}, c_{\mathrm{R}})), \tag{7}$$

where for $l = 0, 1$, $L_{tl}(c_{\mathrm{L}}, c_{\mathrm{R}}) = \frac{1}{B} \sum_{b=1}^B (\mathbb{1}(\mathcal{B}_{tb}(c_{\mathrm{L}}, c_{\mathrm{R}}) = l) - \frac{\sum_{b=1}^B \mathbb{1}(\mathcal{B}_{tb}(c_{\mathrm{L}}, c_{\mathrm{R}})=l)}{B})^2$. Given a desirable pair of hyperparameters $(c_{\mathrm{L}}, c_{\mathrm{R}})$ and $l$ value, the indicator function $\mathbb{1}(\mathcal{B}_{tb}(c_{\mathrm{L}}, c_{\mathrm{R}}) = l)$ is binary, which roughly follows a Bernoulli distribution with probability $\frac{\sum_{b=1}^B \mathbb{1}(\mathcal{B}_{tb}(c_{\mathrm{L}}, c_{\mathrm{R}})=l)}{B}$. Intuitively, the loss function defined in Eq (7) measures the average of squared differences between $\mathbb{1}(\mathcal{B}_{tb}(c_{\mathrm{L}}, c_{\mathrm{R}}) = l)$ and the expected value of Bernoulli$(\frac{\sum_{b=1}^B \mathbb{1}(\mathcal{B}_{tb}(c_{\mathrm{L}}, c_{\mathrm{R}})=l)}{B})$ random variables. We would expect that the optimal pair of hyperparameters $(c_{\mathrm{L}}^t, c_{\mathrm{R}}^t)$ at Stage $t$ minimizes such a loss.

---

**Algorithm 1** Dynamic subgroup identification CARA design

**Stage 1 (Initialization):**
1: Enroll $n_1$ participants, and assign treatments in group $j$ with $e_{1j} = \frac{1}{2}$;
2: Compute $\widehat{\tau}_{1j}$ and $\widehat{\mathbb{V}}_{1j}$;
3: Choosing hyperparameters $(c_{\mathrm{L}}^1, c_{\mathrm{R}}^1)$ using single bootstrap method (see Algorithm 3);
4: Identify tie set and merge tied subgroups with the best subgroup (see Algorithm 2).
    **Stage $t$ (Adaptive treatment allocation revision):**
5: **for** $t \to 2$ to $T$ **do**
6:    With $\widehat{\tau}_{t-1,(j)}$ and $\widehat{\mathbb{V}}_{t-1,(j)}$ estimated using Eq 8, solve the optimization problem in Eq (1) to find $\widehat{e}_{t,(j)}^*$;
7:    Enroll $n_t$ participants and assign treatment with calibrated probability $\widetilde{e}_{t,(j)}^*$ as in Eq (2);
8:    Update $\widehat{\tau}_{t,(j)}$ and $\widehat{\mathbb{V}}_{t,(j)}$ as in Eq (8);
9:    Choosing hyperparameters $(c_{\mathrm{L}}^t, c_{\mathrm{R}}^t)$ using single bootstrap (see Algorithm 3);
10:    Identify tie set and merge tied subgroups with the best subgroup (see Algorithm 2).
11:    Calculate the merged subgroup ATE estimator $\widehat{\tau}_{t,\widehat{\mathcal{T}}_{t1}}$ as in Eq (9), and its variance estimator $\widehat{\mathbb{V}}_{t,\widehat{\mathcal{T}}_{t1}}$ as in Eq (10).
12: **end for**
    **Stage $T$ (Inference):**
13: Identify the best tie set $\widehat{\mathcal{T}}_1$, and construct two-sided confidence intervals for $\widehat{\tau}_{\widehat{\mathcal{T}}_1}$ as in Eq (11).

---

### 4.2 Statistical inference

Our CARA design also enables making valid statistical inference on the estimated best subgroup treatment effect. We highlight two parts of the statistical inference procedure: (1) estimating unknown

parameters based on accrued experimental data while dynamically identifying best subgroups, and (2) constructing valid confidence intervals to confirm the estimated best subgroup treatment effect.

First, based on accumulated experimental data, the subgroup treatment effects and associated variances can be updated as

$$\widehat{\tau}_{t-1,(j)} = \frac{\sum_{s=1}^{t-1} \sum_{i=1}^{n_s} \mathbb{1}_{(X_{is} \in \mathcal{S}_{(j)})} D_{is} Y_{is}}{N_{t-1,(j)}(1)} - \frac{\sum_{s=1}^{t-1} \sum_{i=1}^{n_s} \mathbb{1}_{(X_{is} \in \mathcal{S}_{(j)})} (1 - D_{is}) Y_{is}}{N_{t-1,(j)}(0)}, \tag{8}$$

$$\widehat{\mathbb{V}}_{t-1,(j)}(e_j) = \frac{\sum_{s=1}^{t-1} \sum_{i=1}^{n_s} \mathbb{1}_{(X_{is} \in \mathcal{S}_{(j)})} D_{is} \left( Y_{is} - \bar{Y}_{t-1,(j)}(1) \right)^2}{N_{t-1,(j)}(1)} \left( \frac{e_j \cdot N_{t-1,(j)}}{N_{t-1}} \right)^{-1}$$

$$+ \frac{\sum_{s=1}^{t-1} \sum_{i=1}^{n_s} \mathbb{1}_{(X_{is} \in \mathcal{S}_{(j)})} (1 - D_{is}) \left( Y_{is} - \bar{Y}_{t-1,(j)}(0) \right)^2}{N_{t-1,(j)}(0)} \left( \frac{(1 - e_j) \cdot N_{t-1,(j)}}{N_{t-1}} \right)^{-1},$$

where $N_{t-1,(j)}(1) = \sum_{s=1}^{t-1} \sum_{i=1}^{n_s} \mathbb{1}_{(X_{is} \in \mathcal{S}_{(j)})} D_{is}$, $N_{t-1,(j)}(0) = \sum_{s=1}^{t-1} \sum_{i=1}^{n_s} \mathbb{1}_{(X_{is} \in \mathcal{S}_{(j)})} (1 - D_{is})$, $N_{t-1,(j)} = \sum_{s=1}^{t-1} \sum_{i=1}^{n_s} \mathbb{1}_{(X_{is} \in \mathcal{S}_{(j)})}$, $\bar{Y}_{t-1,(j)}(0) = \sum_{s=1}^{t-1} \sum_{i=1}^{n_s} \mathbb{1}_{(X_{is} \in \mathcal{S}_{(j)})} (1 - D_{is}) Y_{is} / N_{t-1,(j)}(0)$, and $\bar{Y}_{t-1,(j)}(1) = \sum_{s=1}^{t-1} \sum_{i=1}^{n_s} \mathbb{1}_{(X_{is} \in \mathcal{S}_{(j)})} D_{is} Y_{is} / N_{t-1,(j)}(1)$.

Additionally, denote the subgroup proportions as $p_1, \ldots, p_m$. Following dynamic subgroup identification at each stage, we merge all subgroups in the $\widehat{\mathcal{T}}_{t1}$ and estimate the merged best subgroup treatment effect as

$$\widehat{\tau}_{t,\widehat{\mathcal{T}}_{t1}} = \sum_{j \in \widehat{\mathcal{T}}_{t1}} p_j \widehat{\tau}_{tj} \bigg/ \sum_{j \in \widehat{\mathcal{T}}_{t1}} p_j, \tag{9}$$

and estimate the variance of the merged best subgroup treatment effect estimator as

$$\widehat{\mathbb{V}}_{t,\widehat{\mathcal{T}}_{t1}} = \sum_{j \in \widehat{\mathcal{T}}_{t1}} p_j^2 \widehat{\mathbb{V}}_{tj} \bigg/ \left( \sum_{j \in \widehat{\mathcal{T}}_{t1}} p_j \right)^2. \tag{10}$$

In Stage $T$ (line 13), we let $\widehat{\mathcal{T}}_1 := \widehat{\mathcal{T}}_{T1}$, $\widehat{\tau}_{\widehat{\mathcal{T}}_1} := \widehat{\tau}_{T,\widehat{\mathcal{T}}_1}$ and $\widehat{\mathbb{V}}_{\widehat{\mathcal{T}}_1} := \widehat{\mathbb{V}}_{T,\widehat{\mathcal{T}}_1}$. Lastly, to confirm the estimated best subgroup treatment effect, we construct a two-sided level-$\alpha$ confidence interval as

$$\left[ \widehat{\tau}_{\widehat{\mathcal{T}}_1} \pm \Phi^{-1}(1 - \alpha/2) \cdot \sqrt{\widehat{\mathbb{V}}_{\widehat{\mathcal{T}}_1}/N} \right]. \tag{11}$$

---

**Algorithm 2** Dynamic subgroup identification in Stage $t$

---

**Step 1 (Input):**
1: Input $\{\mathcal{H}_s\}_{s=1}^{t}$, $\widehat{\tau}_{tj}$, $\widehat{\mathbb{V}}_{tj}$, and $(c_{\mathtt{L}}^t, c_{\mathtt{R}}^t)$ computed from Algorithm 3.
   **Step $b$ (Bootstrap):**
2: **for** $b \leftarrow 1$ to $B$ **do**
3:     **if** $t = 1$ **then**
4:         Generate $\widehat{\tau}_1^\circ$ from $\mathcal{N}\left( \widehat{\tau}_1, \widehat{\Omega}_n/n_1 \right)$, where $\widehat{\Omega}_n = \mathtt{diag}\left( \widehat{\mathbb{V}}_{11}, \ldots, \widehat{\mathbb{V}}_{1m} \right)$;
5:     **else if** $t > 1$ **then**
6:         Generate $n_s$ resamples randomly with replacement sequentially from each $\mathcal{H}_s$;
7:         Compute $\widehat{\tau}_{tj}^\circ$ as in Eq (8) with the bootstrap samples;
8:     **end if**
9:     Identify the best subgroups $\widehat{\mathcal{T}}_{t1}$ with the bootstrap samples as in Eq (3).
10: **end for**
   **Step $B$ (Output):**
11: Choose $\widehat{\mathcal{T}}_{t1}$ with the highest frequency of occurrence and merge subgroups that belong to $\widehat{\mathcal{T}}_{t1}$.

---

# 5 Theoretical investigation

**Assumption 1** (Regularity conditions). $(Y_{it}(0), Y_{it}(1), X_{it})$ *are independently identically distributed for $i = 1, \ldots, n_t$, $t = 1, \ldots, T$. In addition, $\mathbb{E}[|Y_{it}(d)|]^4 < \infty$, $d \in \{0, 1\}$. Lastly, there exists some $\delta > 0$, such that $\mathbb{V}[Y_{it}(d)|X_{it} \in \mathcal{S}_j] \geq \delta$ for $d \in \{0, 1\}$, $j = 1, \ldots, m$.*

**Assumption 2** (Positivity). *The subgroup proportions $\delta \leq p_1, \ldots, p_m \leq 1 - \delta$, $\delta \in (0, 1/2)$.*

Assumption 1 says that the potential outcomes have bounded moments and have variability in each subgroup. Assumption 2 says that the subgroup proportions are non-zero in the population.

**Theorem 1** (Dynamic best subgroup identification consistency). *Under Assumptions 1 and 2, for $j = 1, \ldots, m$ and for $\varepsilon > 0$, we have*

$$\lim_{N \to \infty} \mathbb{P}\left(|\mathbb{1}(j \in \widehat{\mathcal{T}}_1) - \mathbb{1}(j \in \mathcal{T}_1)| > \varepsilon\right) = 0.$$

Theorem 1 suggests that our dynamic subgroup identification algorithm correctly identifies the best set of subgroups as the sample size tends to infinity.

**Theorem 2** (Design strategy consistency). *Under Assumptions 1 and 2, for $\delta > 0$, as $n_t \to \infty$, $t = 1, \ldots, T$, for the actual treatment allocation, we have*

$$\mathbb{P}(\|\widehat{\boldsymbol{e}}_t - \boldsymbol{e}^*\| \leq \delta) \to 1,$$

where $\boldsymbol{e}^* = (e_1^*, \ldots, e_m^*)$ is the optimal treatment allocation.

Theorem 2 says that the actual treatment allocation under our proposed design strategy converges to the optimal treatment allocation asymptotically.

**Theorem 3** (Asymptotic normality). *Under Assumptions 1 and 2, as $N \to \infty$,*

$$\sqrt{N}(\widehat{\tau}_{\widehat{\mathcal{T}}_1} - \tau_{\mathcal{T}_1}) \to \mathcal{N}\left(0, \mathbb{V}_{\mathcal{T}_1}(e_1^*)\right),$$

*and*

$$\widehat{\mathbb{V}}_{\widehat{\mathcal{T}}_1} - \mathbb{V}_{\mathcal{T}_1}(e_1^*) = O_p(\frac{1}{\sqrt{N}}).$$

Theorem 3 says that the estimated treatment effect of the identified best subgroups converge to a Gaussian distribution asymptotically, and our variance estimator consistently estimates the asymptotic variance. Theorem 3 also verifies the validity of our constructed confidence interval.

# 6 Synthetic real data study

In this section, we investigate the performance of our proposed design strategy for identifying the tie set of best-performing subgroups in a synthetic case study using clinical trial data.

We design our synthetic case study using the dataset from the Mayo Clinic's trial on primary biliary cirrhosis (PBC), containing clinical biomarkers, treatments, and patient outcomes. PBC is a progressive autoimmune liver disease marked by inflammation and damage to the intrahepatic bile ducts. The Mayo Clinic conducted an extensive trial from 1974 to 1984 to assess the effectiveness of D-penicillamine in treating PBC. This dataset includes 424 patients, encompassing both those who were actively enrolled in the trial and additional cases who consented to provide basic measurements [26].

In this case study, we work with a subset ($n = 312$) of patients who participate in the randomized controlled trial. These patients are randomly assigned to one of the two arms: the treatment arm ($n = 158$), who receive D-penicillamine $D = 1$, and the control arm ($n = 154$), who receive placebos $D = 0$. The outcome of interest is the square root of the survival time, defined as the number of days from registration to the earlier death, transplantation, or the time of study analysis. This dataset includes 17 covariates, and we use median imputation to handle missingness in these covariates. We aim to investigate the effectiveness of D-penicillamine in improving liver function and symptoms in five subgroups defined by age (in days): (1) patients with age in $[9, 598, 15, 695]$, (2) age in $(15, 695, 17, 082]$, (3) age in $(17, 082, 20, 440]$, (4) age in $(20, 440, 21, 900]$, (5) age in $(21, 900, 28, 650]$. We generate synthetic experimental data based on the original dataset, which shall be illustrated in the next section.

We generate synthetic data that mimic the original PBC dataset. Denote the subgroup membership for each participant $i$ as $\boldsymbol{S} = (\mathbb{1}_{(X_i \in \mathcal{S}_1)}, \ldots, \mathbb{1}_{(X_i \in \mathcal{S}_5)})^{\mathsf{T}}$. We generate the potential outcome from $Y_i(d)|X_i \in \mathcal{S}_j \sim \mathcal{N}(\mu_{dj}, \sigma_{dj}^2), j = 1, \ldots, 5$, where $\boldsymbol{\mu}_1 = (42.57, 50.44, 44.37, 44.30, 37.71)^{\mathsf{T}}, \boldsymbol{\mu}_0 = (45.34, 39.91, 45.58, 33.42, 39.17)^{\mathsf{T}}, \boldsymbol{\sigma}_1 = (10.85, 12.29, 12.64, 14.28, 14.64)^{\mathsf{T}}, \boldsymbol{\sigma}_0 = (11.50, 15.18, 14.57, 13.09, 15.06)^{\mathsf{T}}$.

The subgroup proportions are $\boldsymbol{p} = (0.28, 0.13, 0.30, 0.11, 0.19)^{\mathsf{T}}$. We denote the true subgroup treatment effects as $\boldsymbol{\tau} = (-2.77, 10.53, -1.21, 10.89, -1.46)^{\mathsf{T}}$. Therefore, Subgroup 2 and Subgroup 4 are the set of best subgroups with a merged average treatment effect of 10.70. The treatment assignment $D_i$ is decided based on different experiment strategies, which shall be discussed later in the section. To generate synthetic data, We mimic CARA experiments where participants are enrolled sequentially across $T$ experimental stages. Here, we set $T = 15$ and $n_t = 400$, for $t = 1, \ldots, T$. All experiments are conducted with an Intel Core i7-11800H CPU and 16 GB of RAM.

**Algorithm 3** Hyperparameter selection for dynamic subgroup identification

**Step 1 (Input):**
1: Input $\{\mathcal{H}_s\}_{s=1}^t$, $\widehat{\tau}_{tj}$, $\widehat{\mathbb{V}}_{tj}$, and $(c_{\mathrm{L}}, c_{\mathrm{R}})$;
2: Compute $\tau_{tj}^*$ as in Eq (4).

**Step $b$ (Bootstrap):**
3: **for** $b \leftarrow 1$ to $B$ **do**
4:   **if** $t = 1$ **then**
5:     Generate $\widehat{\boldsymbol{\tau}}_1^*$ from $\mathcal{N}\left(\boldsymbol{\tau}_1^*, \widehat{\Omega}_n/n_1\right)$, where $\widehat{\Omega}_n = \mathtt{diag}\left(\widehat{\mathbb{V}}_{11}, \ldots, \widehat{\mathbb{V}}_{1m}\right)$;
6:   **else if** $t > 1$ **then**
7:     Generate $n_s$ resamples randomly with replacement sequentially from each $\mathcal{H}_s$ (the same as Algorithm 2 line 6);
8:     Compute $\widehat{\tau}_{tj}^{\circ}$ as in Eq (8) with the bootstrap samples, and then $\widehat{\tau}_{tj}^*$ as in Eq (5);
9:   **end if**
10:   Compute $\widetilde{\tau}_{t,(1)}^*$ as in Eq (6), and $\mathcal{B}_{tb}(c_{\mathrm{L}}, c_{\mathrm{R}})$ as in Eq (7).
11: **end for**

**Step $B$ (Output):**
12: Compute $L_t(c_{\mathrm{L}}, c_{\mathrm{R}})$ as in Eq (7). Choose the pair $(c_{\mathrm{L}}^t, c_{\mathrm{R}}^t)$ that minimizes $L_t(c_{\mathrm{L}}, c_{\mathrm{R}})$.

We compare our proposed design strategy with the complete randomization design and two multi-armed bandit (MAB) algorithms. (1) The complete randomization design refers to a design that fixes $e_{tj} = \frac{1}{2}$ across all experimental stages, $t = 1, \ldots, T, j = 1, \ldots, m$. (2) To customize the MAB algorithms to our setting, for each subgroup, we consider two candidate arms: treatment and control. We set the rewards as the negative asymptotic variance of the treatment effect estimator, i.e., $-\mathbb{V}_j(e_j)$. We consider two MAB algorithms: (a) The $\epsilon$-greedy algorithm, which aims to balance the exploration and the exploitation efforts [37]. Here, we set $\epsilon = 0.1$. (b) The upper confidence bound 1 algorithm balances exploration and exploitation using confidence intervals and chooses the arm that maximizes the upper confidence bound on the estimated reward [4]. When customizing MAB algorithms to our setting, we omit the step of identifying and merging tie sets using conventional methods, such as K-means or agglomerative clustering, due to several challenges: the requirement to predefine the number of clusters, the limited applicability of clustering for a small number of subgroups, and the inconsistency of these methods in effectively merging the best subgroups.

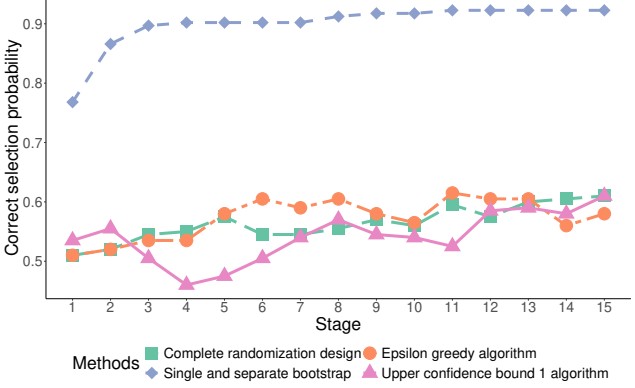

Figure 1: Comparison of the correct selection probability among three conventional methods and our proposed design strategy. "Single and separate bootstrap" refers to our proposed design.

Table 1: Comparison among three conventional methods and our proposed design strategy based on estimated best tie set or subgroup treatment effect (Est), 95% confidence interval (95% CI), $\sqrt{N}$-scaled bias, and standard deviation (SD).

| Method | Est (95% CI) | $\sqrt{N}$Bias | SD |
|---|---|---|---|
| CR | 11.33 (9.30,13.36) | 34.55 | 80.29 |
| $\epsilon$-greedy | 11.26 (9.20,13.31) | 28.84 | 81.19 |
| UCB 1 | 11.34 (9.29,13.38) | 34.82 | 80.85 |
| Proposed | 10.31 (9.28,11.35) | 29.98 | 40.94 |

To evaluate the performance of different design strategies, we assess the effectiveness of each adaptive experiment strategy from two aspects. First, we compare the correct selection probability of identifying the best subgroups. The correction selection probability can be written as $\mathbb{P}(\widehat{\tau}_{\mathcal{T}_1} \geq \max_{j \notin \mathcal{T}_1} \widehat{\tau}_j)$. Second, we compare the 95% confidence interval, $\sqrt{N}$-scaled bias, and standard deviation of the estimated best subgroup treatment effect. In our resampling procedure, we set $B = 2,000$. The synthetic case study results are summarized in Figure 1 and Table 1.

First, from Figure 1, our proposed design strategy shows a higher correct selection probability than the complete randomization design and the MAB algorithms. Additionally, the correct selection probability under our proposed design strategy increases with the number of experimental stages. Specifically, our proposed design has a correct selection probability tending to 1 after 15 experimental stages. We also adopt the normalized mutual information as an additional metric to compare our proposed design and three competing methods in the Appendix (Section K.1), which further confirms that our proposed design strategy outperforms the conventional methods.

Second, from Table 1, we observe that our proposed design strategy has a smaller standard deviation and smaller $\sqrt{N}$-scaled bias, implying that our method is more efficient and less biased. In sum, our proposed adaptive design demonstrates efficient use of experimental data to correctly identify the best-performing subgroups of the three competing methods. We defer additional simulation studies to the Appendix (Section K.1) and an additional synthetic real data study to the Appendix (Section K.3). We also extend our proposed dynamic subgroup identification with CARA to the augmented inverse propensity score weighting (AIPW) estimator. Then we compare our proposed design with AIPW estimator with the three contextual MAB algorithms in the Appendix (Section K.2).

## 7   Discussion

We propose a dynamic subgroup identification method within the CARA design framework that could significantly advance precision medicine. However, we acknowledge some limitations in our approach. We aim to explore and address these challenges in our future research.

First, our method assumes that outcomes in CARA experiments are observed immediately at the end of each stage without delay. This assumption, prevalent in adaptive experiments such as [15] and [49], simplifies the modeling process and facilitates quick adjustments based on the latest data. However, in some practical scenarios, outcomes may be observed with delays, complicating the process of adjusting treatment allocations, as highlighted by [31] and [29]. In future work, we plan to revise our design framework and update our estimators to account for the impact of delayed responses on both treatment effects and variance estimators.

Second, our work addresses scenarios where assigning a treatment is costly, and there is an overall constraint on how many treatments can be deployed. A key challenge arises from the potential misalignment between efficiently estimating the best causal effect and determining the optimal causal decision rule–particularly when the best decision rule is to treat all patients when there is any positive effect. For instance, [9] focuses on ensuring that all individuals who would benefit are accurately assigned to the treatment arm, assuming negligible treatment costs. To maximize the welfare for participating subjects, in future work, we plan to incorporate an "early stopping" step which would not only identify the most effective subgroups but also halt enrollment for subgroups exhibiting significantly adverse treatment effects.

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

# Technical Appendix

## A  Notations

We present a notation table, as shown in Table 2, to illustrate the key symbols and their descriptions used in our covariate-adjusted response-adaptive randomization (CARA) experiment framework.

Table 2: Notation table for the proposed dynamic subgroup identification strategy with CARA design.

| Symbol | Description |
|---|---|
| $T$ | Total number of stages in the experiment. |
| $N$ | Total number of enrolled participants across all stages. |
| $n_{tj}$ | Number of subjects enrolled in subgroup $j$ at Stage $t$. |
| $n_t$ | Number of participants enrolled in Stage $t$. |
| $N_{tj}$ | Cumulative sample size for subgroup $j$ up to Stage $t$. |
| $N_t$ | Cumulative sample size up to Stage $t$. |
| $D_{it}$ | Treatment assignment status of participant $i$ in Stage $t$. |
| $Y_{it}$ | Observed outcome for participant $i$ in Stage $t$. |
| $Y_{it}(d)$ | Potential outcome for participant $i$ if assigned treatment $d$ in Stage $t$. |
| $X_{it}$ | Covariate information for participant $i$ in Stage $t$. |
| $\mathcal{X}$ | Covariate space. |
| $\mathcal{H}_t$ | Historical information up to Stage $t$. |
| $m$ | Total number of subgroups. |
| $m_t^*$ | Number of subgroups after merging the identified tie set in Stage $t-1$. |
| $p_j$ | The $j$-th subgroup proportion. |
| $\widehat{p}_{tj}$ | Estimated subgroup proportion of subgroup $j$ at Stage $t$. |
| $e_{tj}$ | Treatment assignment probability for participants in subgroup $j$ at Stage $t$. |
| $\widehat{e}_{tj}^*$ | Optimal treatment allocation probabilities at Stage $t$. |
| $\mathcal{T}_1$ | Tie set of the highest treatment effect $\tau_1$, containing indices of tied subgroups. |
| $\mathcal{S}_j$ | The $j$-th subgroup. |
| $\mathcal{S}_{\mathcal{T}_1}$ | Subgroups that belong to the best set identified by $\mathcal{T}_1$. |
| $\tau_j$ | Subgroup average treatment effect for subgroup $j$. |
| $\tau_{\mathcal{T}_1}$ | Population treatment effect under the best subgroups in $\mathcal{T}_1$. |

## B  Assumptions, lemmas, and corollaries

Before discussions, we define some additional notations first. For $t = 1, \ldots, T$, $j = 1, \ldots, m$, and $c > 0$, denote

$$\mu_{j,c}(1) = \mathbb{E}[Y_{it}(1)^c | X_{it} \in \mathcal{S}_j], \quad \mu_{j,c}(0) = \mathbb{E}[Y_{it}(0)^c | X_{it} \in \mathcal{S}_j],$$

$$\widehat{\mu}_{tj,c}(1) = \frac{\sum_{s=1}^{t} \sum_{i=1}^{n_s} \mathbb{1}_{(X_{is} \in \mathcal{S}_j)} D_{is} Y_{is}^c}{N_{tj}(1)}, \quad \widehat{\mu}_{tj,c}(0) = \frac{\sum_{s=1}^{t} \sum_{i=1}^{n_s} \mathbb{1}_{(X_{is} \in \mathcal{S}_j)} (1 - D_{is}) Y_{is}^c}{N_{tj}(0)}.$$

**Assumption 3** (Subgroup treatment effects). *For $\delta \in (0, 0.5)$, the asymptotic distance between treatment effects of Subgroup $j$ and Subgroup 1 diverges as $N \to \infty$:*

$$N^\delta \cdot \min_{j \notin \mathcal{T}_1} |\tau_1 - \tau_j| \to \infty, \ \forall j = 1, \ldots, m.$$

**Lemma 1.** *Assume Assumption 1 and 2 holds. Let $p_j = \mathbb{P}(X_{it} \in \mathcal{S}_j | \mathcal{H}_{t-1})$ be the subgroup proportions, and $\widehat{e}_{tj}^* = \mathbb{P}(D_{it} = 1 | X_{it} \in \mathcal{S}_j, \mathcal{H}_{t-1})$ be the treatment probabilities. Assume there exists some $\delta \in (0, 1/2)$ such that for all $j = 1, 2, \ldots, m$ and $t = 1, 2, \ldots T$,*

$$\delta \leq \widehat{e}_{tj}^* \leq 1 - \delta.$$

*Then for any $j = 1, 2, \ldots, m$, $d = 0, 1$, and any $t$ satisfying $n_t \to \infty$,*

$$\widehat{\mu}_{tj,c}(d) = \mu_{j,c}(d) + O_p\left(\frac{1}{\sqrt{N_t}}\right).$$

**Corollary 1.** *Let $\delta \in (0, 1/2)$ be some constant. Assume Assumptions 1 and 2 hold,*

$$\sup_{\delta \leq e \leq 1 - \delta} \left| \widehat{\mathbb{V}}_{tj}(e) - \mathbb{V}_j(e) \right| = O_p\left( \frac{1}{\sqrt{N_t}} \right).$$

**Corollary 2.** *Assume Assumptions 1 and 2 hold. Then*

$$\widehat{\tau}_{tj} - \tau_j = O_p\left( \frac{1}{\sqrt{N_t}} \right).$$

**Theorem 4.** *Under Assumptions 1 and 2, as $N \to \infty$,*

$$\sqrt{N}(\widehat{\tau}_j - \tau_j) \to \mathcal{N}\left(0, \mathbb{V}_j(e_j^*)\right), \quad \widehat{\mathbb{V}}_j - \mathbb{V}_j(e_j^*) = O_p(\frac{1}{\sqrt{N}}).$$

Lemma 1, Corollaries 1 and 2, and Theorem 4 are similar to the theoretical results in [41]. We defer readers to [41] for more technical details.

**Lemma 2.** *Under Assumptions 3, for $j = 1, \ldots, m$, $t = 1, \ldots, T$, any positive constant $C$ and $\delta \in (0, \frac{1}{2})$, the following statement holds*

$$\lim_{N_t \to \infty} \mathbb{P}\left( \left| \widehat{\tau}_{tj}^\circ - \tau_j \right| \geq N_t^{-\delta} \cdot C \right) = 0.$$

**Lemma 3.** *Let $\delta \in (0, 1/2)$ be some constant. Assume Assumptions 1 and 2 hold,*

$$\widehat{\tau}_{t, \widehat{\mathcal{T}}_{t1}} - \tau_{\mathcal{T}_1} = O_p\left( \frac{1}{\sqrt{N_t}} \right), \quad \sup_{\delta \leq e \leq 1 - \delta} \left| \widehat{\mathbb{V}}_{t, \widehat{\mathcal{T}}_{t1}}(e) - \mathbb{V}_{\mathcal{T}_1}(e) \right| = O_p\left( \frac{1}{\sqrt{N_t}} \right).$$

**Lemma 4.** *Assume Assumptions 1 and 2 hold. Let $\mathcal{E}^*$ be the solution(s) to the problem as follows:*

$$\max_{\mathbf{e}} \left\{ \min_{k+1 \leq j \leq m} G(\mathcal{S}_{\mathcal{T}_1}, \mathcal{S}_j; e_1, e_j) = \frac{(\tau_j - \tau_{\mathcal{T}_1})^2}{2\left(\mathbb{V}_{\mathcal{T}_1}(e_1) + \mathbb{V}_j(e_j)\right)} \right\},$$

$$s.t. \sum_{j=k+1}^{m} p_j e_j \leq c_1, \quad c_2 < e_j < 1 - c_2, \ j = 1, \ldots, m,$$

*where $c_1 \in (0, 1)$, $c_2 \in (0, 1/2)$, $\mathcal{T}_1 = \{1, \ldots, k\}$, $1 \leq k < m$, and $\mathbb{V}_{\mathcal{T}_1}$ denotes the variance of the top subgroups. Let $\widehat{\mathcal{E}}^*$ be the optimized treatment allocations solved from the sample analog of this problem. Then for any $\delta > 0$, and any $t$ satisfying $N_{t-1} \to \infty$,*

$$\mathbb{P}\left( \sup_{\widehat{\mathbf{e}}_t^* \in \widehat{\mathcal{E}}^*} \inf_{\mathbf{e}^* \in \mathcal{E}^*} \left\| \widehat{\mathbf{e}}_t^* - \mathbf{e}^* \right\| \leq \delta \right) \to 1.$$

## C Optimization problem objective function derivations

*Proof.* Because maximizing the correct selection probability is equivalent to minimizing the incorrect selection probability, we formulate the incorrect selection probability as $\mathbb{P}\left( \widehat{\tau}_{\mathcal{T}_1} \leq \max_{j \notin \mathcal{T}_1} \widehat{\tau}_j \right)$ and denote the cardinality of $\mathcal{T}_1$ as $k$, $1 \leq k \leq m - 1$, where $m$ denotes the total number of subgroups. The incorrect selection probability can be bounded as

$$\max_{j \notin \mathcal{T}_1} \mathbb{P}\left( \widehat{\tau}_{\mathcal{T}_1} \leq \widehat{\tau}_j \right) \leq \mathbb{P}\left( \widehat{\tau}_{\mathcal{T}_1} \leq \max_{j \notin \mathcal{T}_1} \widehat{\tau}_j \right) \leq (m - k) \cdot \max_{j \notin \mathcal{T}_1} \mathbb{P}\left( \widehat{\tau}_{\mathcal{T}_1} \leq \widehat{\tau}_j \right),$$

where the inequality follows from the union bound. For both sides, take the logarithm and divide by $N$,

$$\frac{1}{N} \log \max_{j \notin \mathcal{T}_1} \left( 1 - \mathbb{P}\left( \widehat{\tau}_{\mathcal{T}_1} \geq \widehat{\tau}_j \right) \right) \leq \frac{1}{N} \log (m - k) + \frac{1}{N} \log \left( \max_{j \notin \mathcal{T}_1} \left( 1 - \mathbb{P}\left( \widehat{\tau}_{\mathcal{T}_1} \geq \widehat{\tau}_j \right) \right) \right).$$

Additionally, by the large deviation theory [8], there exists a rate function $G(\mathcal{S}_{\mathcal{T}_1}, \mathcal{S}_j; e_1, e_j)$, such that

$$\lim_{N \to \infty} \frac{1}{N} \log \left( 1 - \mathbb{P}\left( \widehat{\tau}_{\mathcal{T}_1} \geq \widehat{\tau}_j \right) \right) = -G(\mathcal{S}_{\mathcal{T}_1}, \mathcal{S}_j; e_1, e_j), \ j \notin \mathcal{T}_1.$$

Therefore, $\max_{j \notin \mathcal{T}_1} \left(1 - \mathbb{P}(\widehat{\tau}_{\mathcal{T}_1} \geq \widehat{\tau}_j)\right)$ is equivalent to $\exp\left(-\min_{j \notin \mathcal{T}_1} G(\mathcal{S}_{\mathcal{T}_1}, \mathcal{S}_j; e_1, e_j)\right)$. We obtain

$$-\min_{j \notin \mathcal{T}_1} G(\mathcal{S}_{\mathcal{T}_1}, \mathcal{S}_j; e_1, e_j) \leq \frac{1}{N} \log\left(1 - \mathbb{P}\left(\widehat{\tau}_{\mathcal{T}_1} \geq \max_{j \notin \mathcal{T}_1} \widehat{\tau}_j\right)\right)$$

$$\leq \frac{\log(m-k)}{N} - \min_{j \notin \mathcal{T}_1} G(\mathcal{S}_{\mathcal{T}_1}, \mathcal{S}_j; e_1, e_j).$$

As a result, based on the Gartner-Ellis Theorem [8, ch.2.3], when the sample size tends to infinity, we are able to obtain:

$$\lim_{N \to \infty} \frac{1}{N} \log\left(1 - \mathbb{P}\left(\widehat{\tau}_{\mathcal{T}_1} \geq \max_{j \notin \mathcal{T}_1} \widehat{\tau}_j\right)\right) = -\min_{j \notin \mathcal{T}_1} G(\mathcal{S}_1, \mathcal{S}_j; e_1, e_j),$$

$$G(\mathcal{S}_{\mathcal{T}_1}, \mathcal{S}_j; e_1, e_j) = \frac{(\tau_j - \tau_{\mathcal{T}_1})^2}{2(\mathbb{V}_{\mathcal{T}_1}(e_1) + \mathbb{V}_j(e_j))}.$$

$\square$

## D    Proof of Lemma 2

*Proof.* Note that

$$\sqrt{N_t}\left(\widehat{\tau}_{tj}^{\circ} - \tau_j\right) = \sqrt{N_t}\left(\widehat{\tau}_{tj} - \tau_j\right) + \sqrt{N_t}\left(\widehat{\tau}_{tj}^{\circ} - \widehat{\tau}_{tj}\right).$$

As for the first term $\sqrt{N_t}\left(\widehat{\tau}_{tj} - \tau_j\right)$, by Corollary 2, we have

$$\sqrt{N_t}\left(\widehat{\tau}_{tj} - \tau_j\right) = O_p(1).$$

As for the second term $\sqrt{N_t}\left(\widehat{\tau}_{tj}^{\circ} - \widehat{\tau}_{tj}\right)$, we consider two situations under $t = 1$ and $t > 1$ separately. When $t = 1$, $\widehat{\tau}_{1j}^{\circ}$ is generated from $\mathcal{N}\left(\tau_{1j}^{\circ}, \widehat{\mathbb{V}}_{1j}/N_1\right)$, so we have

$$\sqrt{N_1}\left(\widehat{\tau}_{1j}^{\circ} - \tau_{1j}^{\circ}\right) = O_p(1).$$

When $t > 1$, we compute $\widehat{\tau}_{tj}^{\circ}$ the same way as $\widehat{\tau}_{tj}$ with the bootstrap samples instead, so $\widehat{\tau}_{tj}^{\circ}$ and $\widehat{\tau}_{tj}$ are consistent and we have

$$\sqrt{N_t}\left(\widehat{\tau}_{tj}^{\circ} - \widehat{\tau}_{tj}\right) = O_p(1).$$

Therefore, we obtain for $t \geq 1$, $\sqrt{N_t}\left(\widehat{\tau}_{tj}^{\circ} - \tau_j\right) = O_p(1)$, i.e., for any given $\varepsilon > 0$, there exists an $M$, such that

$$\mathbb{P}\left(\sqrt{N_t}|\widehat{\tau}_{tj}^{\circ} - \tau_j| > M\right) < \varepsilon.$$

Then for any $N_t$ such that $N_t > \left(\frac{M}{C}\right)^{\frac{1}{\frac{1}{2} - \delta}}$, we have

$$\mathbb{P}\left(|\widehat{\tau}_{tj}^{\circ} - \tau_j| > N_t^{-\delta} \cdot C\right) < \varepsilon,$$

and thus

$$\limsup_{n \to \infty} \mathbb{P}\left(|\widehat{\tau}_{tj}^{\circ} - \tau_j| \geq N_t^{-\delta} \cdot C\right) < \varepsilon.$$

Note that the above inequality holds for arbitrary $\varepsilon > 0$. Hence, we have

$$\limsup_{n \to \infty} \mathbb{P}\left(|\widehat{\tau}_{tj}^{\circ} - \tau_j| \geq N_t^{-\delta} \cdot C\right) = 0,$$

which completes the proof.

$\square$

# E   Proof of Theorem 1

*Proof.* Recall our definition of the tie set of best-performing subgroups:

$$\mathcal{T}_1 = \{j : |\tau_1 - \tau_j| = o(N^{-1/2}),\ j = 1, \ldots, m\}.$$

This suggests that $\forall j \in \mathcal{T}_1$, there exists a sequence $\delta_n \to 0$ as $N \to 0$, such that

$$\tau_j = \tau_1 + N^{-\frac{1}{2}} \cdot \delta_n,\ \forall j \in \mathcal{T}_1.$$

We further define a set of subgroups with treatment effects smaller than those in the set $\mathcal{T}_1$:

$$\mathcal{T}_1^{\mathrm{L}} = \{j : \tau_j < \min_{m \in \mathcal{T}_1}\{\tau_m\},\ j = 1, \ldots, m\}.$$

As for the estimated near tie set of $\tau_1$ in Stage $t$ for $t = 1, \ldots, T$, we have for any $j \in \widehat{\mathcal{T}}_{t1}$, and a pair of hyperparameters $(c_{\mathrm{L}}^t, c_{\mathrm{R}}^t)$ that

$$-c_{\mathrm{L}}^t \cdot N_t^{-\delta} \cdot \widehat{\mathbb{V}}_{t,(1)}^{\delta} \leq (\widehat{\tau}_{tj}^{\circ} - \widehat{\tau}_{t,(1)}^{\circ}) \leq c_{\mathrm{R}}^t \cdot N_t^{-\delta} \cdot \widehat{\mathbb{V}}_{t,(1)}^{\delta},$$

where $\delta = 0.25$. Thus, there exists a positive constant $C$ such that

$$\frac{|\widehat{\tau}_{tj}^{\circ} - \widehat{\tau}_{t,(1)}^{\circ}|}{N_t^{-\delta}} < C,\ \forall j \in \widehat{\mathcal{T}}_{t1}.$$

Similar to the proof of Lemma 1 in [42], our proof is composed of the following three steps:

**Step 1.** We first show that the subgroup with the largest treatment effect in the resampled statistics falls into the set $\mathcal{T}_1$ with high probability, that is

$$\lim_{N_t \to \infty} \mathbb{P}\left(\breve{1}_t \in \mathcal{T}_1\right) = 1, \tag{12}$$

where $\breve{1}_t = \sum_{j=1}^{m_t^*} j \cdot \mathbb{1}(\widehat{\tau}_{tj}^{\circ} = \widehat{\tau}_{t,(1)}^{\circ})$ for $t = 1, \ldots, T$.

Because $\widehat{\tau}_{t,\breve{1}_t}^{\circ} \in [\min_{j \in \mathcal{T}_1} \widehat{\tau}_{tj}^{\circ}, \max_{j \in \mathcal{T}_1} \widehat{\tau}_{tj}^{\circ}]$ by definition, coupled with the fact that

$$\left\{\max_{k \in \mathcal{T}_1^{\mathrm{L}}} \widehat{\tau}_{tk}^{\circ} < \min_{j \in \mathcal{T}_1} \widehat{\tau}_{tj}^{\circ} \leq \max_{j \in \mathcal{T}_1} \widehat{\tau}_{tj}^{\circ}\right\} \subset \left(\breve{1}_t \in \mathcal{T}_1\right).$$

Under Assumption 3, by Lemma 2, for any $k \in \mathcal{T}_1^{\mathrm{L}}$ and $j \in \mathcal{T}_1$, we have

$$\lim_{N_t \to \infty} \mathbb{P}\left(\widehat{\tau}_{tk}^{\circ} < \widehat{\tau}_{tj}^{\circ}\right) = 1,$$

so we can obtain $\lim_{N_t \to \infty} \mathbb{P}\left(\max_{k \in \mathcal{T}_1^{\mathrm{L}}} \widehat{\tau}_{tk}^{\circ} < \min_{j \in \mathcal{T}_1} \widehat{\tau}_{tj}^{\circ} \leq \max_{j \in \mathcal{T}_1} \widehat{\tau}_{tj}^{\circ}\right) = 1$. Thus we have shown that $\lim_{N_t \to \infty} \mathbb{P}\left(\breve{1}_t \in \mathcal{T}_1\right) = 1$.

**Step 2.** We then show, for $j \notin \mathcal{T}_1$, and $t = 1, \ldots, T$,

$$\lim_{N_t \to \infty} \mathbb{P}\left(\mathbb{1}(j \in \widehat{\mathcal{T}}_{t1}) > \varepsilon,\ j \notin \mathcal{T}_1\right) = 0. \tag{13}$$

For any $\varepsilon > 0$ and $j \notin \mathcal{T}_1$, we have the following holds

$$\mathbb{P}\left(\left|\mathbb{1}(j \in \widehat{\mathcal{T}}_{t1})\right| > \varepsilon\right)$$

$$= \mathbb{P}\left(\left|\mathbb{1}(j \in \widehat{\mathcal{T}}_{t1})\right| > \varepsilon | j \in \widehat{\mathcal{T}}_{t1}\right) \cdot \mathbb{P}\left(j \in \widehat{\mathcal{T}}_{t1}\right)$$

$$\quad + \mathbb{P}\left(\left|\mathbb{1}(j \in \widehat{\mathcal{T}}_{t1})\right| > \varepsilon | j \notin \widehat{\mathcal{T}}_{t1}\right) \cdot \mathbb{P}\left(j \notin \widehat{\mathcal{T}}_{t1}\right)$$

$$\leq \mathbb{P}\left(j \in \widehat{\mathcal{T}}_{t1}\right)$$

$$\overset{\mathrm{Def}}{=} \mathbb{P}\left(\frac{|\widehat{\tau}_{tj}^{\circ} - \widehat{\tau}_{t,(1)}^{\circ}|}{N_t^{-\delta}} < C\right)$$

$$=\mathbb{P}\left(\frac{|\widehat{\tau}^\circ_{tj} - \widehat{\tau}^\circ_{t,\check{1}}|}{N_t^{-\delta}} < C\right)$$

$$=\mathbb{P}\left(|\widehat{\tau}^\circ_{tj} - \widehat{\tau}^\circ_{t,\check{1}}| < N_t^{-\delta} \cdot C\right)$$

$$=\mathbb{P}\left(|\left(\widehat{\tau}^\circ_{tj} - \tau_j\right) - \left(\widehat{\tau}^\circ_{t,\check{1}} - \tau_{\check{1}}\right) + \left(\tau_j - \tau_{t,\check{1}}\right)| < N_t^{-\delta} \cdot C\right)$$

$$\leq\mathbb{P}\left(N_t^\delta |\tau_j - \tau_{\check{1}}| - N_t^\delta |\widehat{\tau}^\circ_{tj} - \tau_j| - N_t^\delta \left|\widehat{\tau}^\circ_{t,\check{1}} - \tau_{\check{1}}\right| < C\right)$$

$$=\mathbb{P}\left(N_t^\delta |\widehat{\tau}^\circ_{tj} - \tau_j| + N_t^\delta \left|\widehat{\tau}^\circ_{t,\check{1}} - \tau_{\check{1}}\right| > N_t^\delta |\tau_j - \tau_{\check{1}}| + C\right).$$

By definition, for $j \notin \mathcal{T}_1$,

$$\mathbb{P}\left(N_t^\delta |\tau_j - \tau_{\check{1}}| < C\right) \leq \mathbb{P}\left(N_t^\delta |\tau_j - \tau_{\check{1}}| < C, \check{1} \in \mathcal{T}_1\right) + \mathbb{P}\left(\check{1} \notin \mathcal{T}_1\right)$$
$$\leq \max_{k \in \mathcal{T}_1} \mathbb{P}\left(N_t^\delta |\tau_j - \tau_k| < C, k \in \mathcal{T}_1\right) + \mathbb{P}\left(\check{1} \notin \mathcal{T}_1\right).$$

Under Assumption 3, Lemma 2 and the conclusion in Eq (12) in Step 1 suggest that by letting $N_t \to \infty$ on both sides, we have the above probability converges to zero. Based on above derivation, we have shown that $\lim_{N_t\to\infty} \mathbb{P}\left(\mathbb{1}(j \in \widehat{\mathcal{T}}_{t1}) > \varepsilon, j \notin \mathcal{T}_1\right) = 0$.

**Step 3.** We are left to prove that for all $j \in \mathcal{T}_1$ and $t = 1, \ldots, T$, the following holds for all $\varepsilon > 0$:
$$\lim_{N_t\to\infty} \mathbb{P}\left(\left|\mathbb{1}(j \in \widehat{\mathcal{T}}_{t1}) - 1\right| > \varepsilon\right) = 0.$$

Following similar arguments, for a positive constant $C$, we have for $k, j \in \mathcal{T}_1$, the following statement holds:

$$\mathbb{P}\left(\left|\mathbb{1}(j \in \widehat{\mathcal{T}}_{t1}) - 1\right| > \varepsilon\right)$$

$$=\mathbb{P}\left(\left|\mathbb{1}(j \in \widehat{\mathcal{T}}_{t1}) - 1\right| > \varepsilon | j \in \widehat{\mathcal{T}}_{t1}\right) \cdot \mathbb{P}\left(j \in \widehat{\mathcal{T}}_{t1}\right)$$

$$+ \mathbb{P}\left(\left|\mathbb{1}(j \in \widehat{\mathcal{T}}_{t1}) - 1\right| > \varepsilon | j \notin \widehat{\mathcal{T}}_{t1}\right) \cdot \mathbb{P}\left(j \notin \widehat{\mathcal{T}}_{t1}\right)$$

$$\leq\mathbb{P}\left(j \notin \widehat{\mathcal{T}}_{t1}\right)$$

$$\overset{\text{Def}}{=}\mathbb{P}\left(\frac{|\widehat{\tau}^\circ_{tj} - \widehat{\tau}^\circ_{t,(1)}|}{N_t^{-\delta}} \geq C\right)$$

$$=\mathbb{P}\left(\frac{|\widehat{\tau}^\circ_{tj} - \widehat{\tau}^\circ_{t,\check{1}}|}{N_t^{-\delta}} \geq C\right)$$

$$=\mathbb{P}\left(|\left(\widehat{\tau}^\circ_{tj} - \tau_j\right) - \left(\widehat{\tau}^\circ_{t,\check{1}} - \tau_{\check{1}}\right) + \left(\tau_j - \tau_{\check{1}}\right)| \geq N_t^{-\delta} \cdot C\right)$$

$$\leq\mathbb{P}\left(|\widehat{\tau}^\circ_{tj} - \tau_j| + |\tau_{tj} - \tau_{\check{1}}| + \left|\widehat{\tau}^\circ_{t,\check{1}} - \tau_{\check{1}}\right| \geq N_t^{-\delta} \cdot C\right)$$

$$\leq\mathbb{P}\left(N_t^{\frac{1}{2}} |\widehat{\tau}^\circ_{tj} - \tau_j| + N_t^{\frac{1}{2}} |\tau_j - \tau_{\check{1}}| + N_t^{\frac{1}{2}} \left|\widehat{\tau}^\circ_{t,\check{1}} - \tau_{\check{1}}\right| \geq N_t^{\frac{1}{2}-\delta} \cdot C\right).$$

By definition of the near-tie set, for $j \in \mathcal{T}_1$, we have

$$\mathbb{P}\left(N_t^{\frac{1}{2}} |\tau_j - \tau_{\check{1}}| < C\right) \leq \mathbb{P}\left(N_t^\delta |\tau_j - \tau_{\check{1}}| < C, \check{1} \in \mathcal{T}_1\right) + \mathbb{P}\left(\check{1} \notin \mathcal{T}_1\right)$$
$$= \max_{k \in \mathcal{T}_1} \mathbb{P}\left(N_t^\delta |\tau_j - \tau_k| < C, k \in \mathcal{T}_1\right) + \mathbb{P}\left(\check{1} \notin \mathcal{T}_1\right)$$

Again, under Assumption 3, Lemma 2 and the conclusion in Eq (13) we have derived in Step 1, by letting $N_t \to \infty$ on both side, we have the above probability converges to 1. According to above discussion, we have shown that $\lim_{N_t\to\infty} \mathbb{P}\left(\left|\mathbb{1}(j \in \widehat{\mathcal{T}}_{t1}) - 1\right| > \varepsilon\right) = 0$.

Combining the results obtained in the aforementioned three steps, we have for $j = 1, \ldots, m_t^*$ and for $\varepsilon > 0$, the following holds:
$$\lim_{N_t\to\infty} \mathbb{P}\left(|\mathbb{1}(j \in \widehat{\mathcal{T}}_{t1}) - \mathbb{1}(j \in \mathcal{T}_1)| > \varepsilon\right) = 0, \forall t = 1, \ldots, T.$$

Then it is straightforward to get the final results by taking $t = T$. $\qquad\square$

## F   Proof of Lemma 3

*Proof.* Note that

$$\widehat{\tau}_{t,\widehat{\mathcal{T}}_{t1}} - \tau_{\mathcal{T}_1} = (\widehat{\tau}_{t,\widehat{\mathcal{T}}_{t1}} - \widehat{\tau}_{t,\mathcal{T}_1}) + (\widehat{\tau}_{t,\mathcal{T}_1} - \tau_{\mathcal{T}_1}),$$

$$\widehat{\mathbb{V}}_{t,\widehat{\mathcal{T}}_{t1}}(e) - \mathbb{V}_{\mathcal{T}_1}(e) = (\widehat{\mathbb{V}}_{t,\widehat{\mathcal{T}}_{t1}}(e) - \widehat{\mathbb{V}}_{t,\mathcal{T}_1}(e)) + (\widehat{\mathbb{V}}_{t,\mathcal{T}_1}(e) - \mathbb{V}_{\mathcal{T}_1}(e)).$$

Thus this lemma follows directly from Corollary 1, Corollary 2 and Theorem 1. □

## G   Proof of Lemma 4

*Proof.* Denote the oracle objective function as

$$f(\boldsymbol{e}) = \min_{|\mathcal{T}_1|+1 \leq j \leq m} \frac{(\tau_j - \tau_{\mathcal{T}_1})^2}{\mathbb{V}_{\mathcal{T}_1}(e_1) + \mathbb{V}_j(e_j)} = \min_{|\mathcal{T}_1|+1 \leq j \leq m} \frac{(\tau_{(j-|\mathcal{T}_1|+1)} - \tau_{\mathcal{T}_1})^2}{\mathbb{V}_{\mathcal{T}_1}(e_1) + \mathbb{V}_{(j-|\mathcal{T}_1|+1)}(e_j)},$$

where the second equality follows from regarding the tie set $\mathcal{T}_1$ as a whole after merging the data, and then $\mathcal{T}_1$ ranks 1 and other subgroups rank from 2 to $m - |\mathcal{T}_1| + 1$. Then, the original optimization problem can be written as

$$\max_{\boldsymbol{e}} \ f(\boldsymbol{e}), \qquad \text{s.t.} \ \sum_{j=|\mathcal{T}_1|+1}^{m} p_j e_j \leq c_1, \quad c_2 \leq e_j \leq 1 - c_2, \ j = |\mathcal{T}_1| + 1, \ldots, m.$$

For notational simplicity, let $\mathcal{E}$ be the set of candidate allocation rules (i.e., all allocation rules satisfying the constraints).

Similarly, let $\widehat{f}_t(\boldsymbol{e})$ be the estimated objective function, that is, with $m_t^* = m - |\widehat{\mathcal{T}}_{t-1,1}| + 1$,

$$\widehat{f}_t(\boldsymbol{e}) = \min_{2 \leq j \leq m_t^*} \frac{(\widehat{\tau}_{t-1,(j)} - \widehat{\tau}_{t-1,\widehat{\mathcal{T}}_{t-1,1}})^2}{\widehat{\mathbb{V}}_{t-1,\widehat{\mathcal{T}}_{t-1,1}}(e_1) + \widehat{\mathbb{V}}_{t-1,(j)}(e_j)}$$

$$= \min_{|\widehat{\mathcal{T}}_{t-1,1}|+1 \leq j \leq m} \frac{(\widehat{\tau}_{t-1,(j-|\widehat{\mathcal{T}}_{t-1,1}|+1)} - \widehat{\tau}_{t-1,\widehat{\mathcal{T}}_{t-1,1}})^2}{\widehat{\mathbb{V}}_{t-1,\widehat{\mathcal{T}}_{t-1,1}}(e_1) + \widehat{\mathbb{V}}_{t-1,(j-|\widehat{\mathcal{T}}_{t-1,1}|+1)}(e_j)}.$$

We solve the following optimization problem:

$$\max_{\boldsymbol{e}} \ \widehat{f}_t(\boldsymbol{e}), \qquad \text{s.t.} \ \sum_{j=|\widehat{\mathcal{T}}_{t-1,1}|+1}^{m} p_j e_j \leq c_1, \quad c_2 \leq e_j \leq 1 - c_2, \ j = |\widehat{\mathcal{T}}_{t-1,1}| + 1, \ldots, m.$$

We denote the set of maximizers to the above optimization problem as $\widehat{\mathcal{E}}^*$. Lastly, let the $\delta$-enlargement of $\mathcal{E}^*$ be $\mathcal{E}^* + B_\delta$, where $B_\delta$ is the Euclidean ball centered at the origin with radius $\delta$: $\mathcal{E}^* + B_\delta = \{\boldsymbol{e} + \mathbf{u} : \boldsymbol{e} \in \mathcal{E}^*, \|\mathbf{u}\| = \delta\}$. Finally, we notice that $\sup_{\boldsymbol{e} \in \mathcal{E}} |\widehat{f}_t(\boldsymbol{e}) - f(\boldsymbol{e})| = o_p(1)$ due to Lemma 3, and Theorem 1. By the same proof of Lemma C.2 in [41], we have $\widehat{\mathcal{E}}^* \subseteq \mathcal{E}^* + B_\delta$, which closes the proof. □

## H   Proof of Theorem 2

*Proof.* Adopting the same proof of Theorem 1 in [41], we obtain that

$$\widehat{e}_j = \frac{\sum_{t=1}^{T} \sum_{i=1}^{n_t} \mathbb{1}_{(X_{it} \in \mathcal{S}_j)}(D_{it} - \widehat{e}_{tj}^*)}{\sum_{t=1}^{T} \sum_{i=1}^{n_t} \mathbb{1}_{(X_{it} \in \mathcal{S}_j)}} + \frac{\sum_{t=1}^{T} \sum_{i=1}^{n_t} \mathbb{1}_{(X_{it} \in \mathcal{S}_j)} \widehat{e}_{tj}^*}{\sum_{t=1}^{T} \sum_{i=1}^{n_t} \mathbb{1}_{(X_{it} \in \mathcal{S}_j)}}$$

$$= O_p\left(\frac{1}{\sqrt{N}}\right) + \left(1 + o_p(1)\right) \frac{\sum_{t=1}^{T} n_t \widehat{e}_{tj}^*}{N}.$$

From Lemma 4, one has

$$\sup_{2 \leq t \leq T} |\widehat{e}_{tj}^* - e_j^*| = o_p(1).$$

Then with the calibration procedure presented in Section 4.1, by replacing $\widehat{e}_{tj}^*$ with $\widetilde{e}_{tj}$, we have that

$$
\begin{aligned}
\frac{\sum_{t=1}^{T} n_t \widetilde{e}_{tj}}{N} &= \sum_{t=1}^{T} \frac{n_t}{N} \frac{\left(\widehat{e}_{tj}^*(n_{tj} + N_{t-1,j})\right) - N_{t-1,j}(1)}{n_{tj}} \\
&= \sum_{t=1}^{T} \frac{n_t}{N} \widehat{e}_{tj}^* + \sum_{t=1}^{T} \frac{n_t}{N} \frac{\widehat{e}_{tj}^* N_{t-1,j} - N_{t-1,j}(1)}{n_{tj}} \\
&= e_j^* - \sum_{t=1}^{T} \frac{1}{N p_j} \sum_{s=1}^{t-1} \sum_{i=1}^{n_s} \mathbb{1}_{(X_{is} \in \mathcal{S}_j)} \left(D_{is} - \widehat{e}_{tj}^*\right) + o_p(1) \\
&= e_j^* + o_p(1),
\end{aligned}
$$

where the second term in the last equality has the order of $O_p(1/\sqrt{N})$ with variance calculation. Thus, we complete the proof. $\square$

# I  Proof of Theorem 3

*Proof.* Note that

$$
\widehat{\tau}_{\widehat{\mathcal{T}}_1} - \tau_{\mathcal{T}_1} = (\widehat{\tau}_{\widehat{\mathcal{T}}_1} - \widehat{\tau}_{\mathcal{T}_1}) + (\widehat{\tau}_{\mathcal{T}_1} - \tau_{\mathcal{T}_1}),
$$

and

$$
\begin{aligned}
\widehat{\tau}_{\mathcal{T}_1} - \tau_{\mathcal{T}_1} &= \sum_{j \in \mathcal{T}_1} p_j \widehat{\tau}_j \Big/ \sum_{j \in \mathcal{T}_1} p_j - \sum_{j \in \mathcal{T}_1} p_j \tau_j \Big/ \sum_{j \in \mathcal{T}_1} p_j \\
&= \sum_{j \in \mathcal{T}_1} p_j (\widehat{\tau}_j - \tau_j) \Big/ \sum_{j \in \mathcal{T}_1} p_j.
\end{aligned}
$$

Applying the result presented in Theorem 4,

$$
\sqrt{N}(\widehat{\tau}_j - \tau_j) \xrightarrow{\mathcal{D}} \mathcal{N}\left(0, \mathbb{V}_j(e_j^*)\right),
$$

we have that

$$
\sqrt{N}(\widehat{\tau}_{\mathcal{T}_1} - \tau_{\mathcal{T}_1}) \xrightarrow{\mathcal{D}} \mathcal{N}\left(0, \mathbb{V}_{\mathcal{T}_1}(e_1^*)\right),
$$

where $\mathbb{V}_{\mathcal{T}_1}(e_1^*) = \sum_{j \in \mathcal{T}_1} p_j^2 \mathbb{V}_j\left(e_j^*\right) \Big/ \left(\sum_{j \in \mathcal{T}_1} p_j\right)^2$. By Theorem 1, we also have $\widehat{\tau}_{\widehat{\mathcal{T}}_1} - \widehat{\tau}_{\mathcal{T}_1} \to_p 0$. Thus by Slutsky's theorem,

$$
\sqrt{N}(\widehat{\tau}_{\widehat{\mathcal{T}}_1} - \tau_{\mathcal{T}_1}) \xrightarrow{\mathcal{D}} \mathcal{N}\left(0, \mathbb{V}_{\mathcal{T}_1}(e_1^*)\right).
$$

Consistency of the variance estimator follows from Theorem 1 and Lemma 1, which is similar to the proof of Lemma 3. $\square$

# J  Additional algorithms

In this section, we provide another data-adaptive algorithm for choosing hyperparameters for Stage 1 and another bootstrap-based merging data bootstrap for later stages.

We present a double bootstrap method for choosing hyperparameters in Algorithm 4 as an alternative to Algorithm 3.

---

**Algorithm 4** Hyperparameter selection in Stage 1

---

**Step 1 (Input):**
1: Input $\mathcal{H}_1, \widehat{\tau}_{1j}, \widehat{\mathbb{V}}_{1j}$, and $(c_{\mathtt{L}}, c_{\mathtt{R}})$.
2: Compute $\tau^*_{1j}$ as in Eq (4).
    **Step $b$ (Bootstrap):**
3: **for** $b \leftarrow 1$ to $B$ **do**
4:    Generate $\widehat{\tau}^*_1$ from $\mathcal{N}\left(\boldsymbol{\tau}^*_1, \widehat{\Omega}_n/n_1\right)$, where $\widehat{\Omega}_n = \mathtt{diag}\left(\widehat{\mathbb{V}}_{11}, \ldots, \widehat{\mathbb{V}}_{1m}\right)$;
5:    **for** $r \leftarrow 1$ to $R$ **do**
6:        Generate $\widehat{\boldsymbol{\tau}}^{**}_1$ from $\mathcal{N}\left(\widehat{\boldsymbol{\tau}}^*_1, \widehat{\Omega}_n/n_1\right)$;
7:        Compute $\widetilde{\tau}^{**}_{1,(1)}$ as in Eq (6);
8:    **end for**
9:    Record $\mathcal{B}_{1,(b)}(c_{\mathtt{L}}, c_{\mathtt{R}})$ as in Eq (14).
10: **end for**
    **Step $B$ (Output):**
11: Compute $L_1(c_{\mathtt{L}}, c_{\mathtt{R}})$ as in Eq (14). Choose the pair $(c^1_{\mathtt{L}}, c^1_{\mathtt{R}})$ that minimizes $L_1(c_{\mathtt{L}}, c_{\mathtt{R}})$.

---

Here, we define the loss function

$$\mathcal{B}_{1,(b)}(c_{\mathtt{L}}, c_{\mathtt{R}}) = \frac{1}{R}\sum_{r=1}^{R}\mathbb{1}\left(\widetilde{\tau}^{**,r}_{(1)} \leq \tau^{*,r}_{(1)}\right), \quad L_1(c_{\mathtt{L}}, c_{\mathtt{R}}) = \frac{1}{B}\sum_{b=1}^{B}\left(\mathcal{B}_{1,(b)}(c_{\mathtt{L}}, c_{\mathtt{R}}) - \frac{b}{B+1}\right)^2,$$
(14)

where $\mathcal{B}_{1,(b)}(c_{\mathtt{L}}, c_{\mathtt{R}})$ is the $b$-th smallest statistics in $\mathcal{B}_{11}(c_{\mathtt{L}}, c_{\mathtt{R}}), \ldots, \mathcal{B}_{1B}(c_{\mathtt{L}}, c_{\mathtt{R}})$.

We could know that $\mathbb{P}(\widetilde{\tau}^*_{1,(1)} \leq \tau_1 | (Y_{is}, D_{is}, X_{is})_{i=1}^{n_1})$ roughly follows Unif(0,1) when the sample size $n_1$ is large. Given a desirable pair of hyperparameters $(c_{\mathtt{L}}, c_{\mathtt{R}})$, we would thus expect that $\mathcal{B}_{1,(1)}(c_{\mathtt{L}}, c_{\mathtt{R}}), \ldots, \mathcal{B}_{1,(B)}(c_{\mathtt{L}}, c_{\mathtt{R}})$ share a similar distribution with the ordered statistics of i.i.d. Unif(0, 1) random variables. The loss function defined in Eq (14) measures the average of squared differences between $\mathcal{B}_{1,(b)}(c_{\mathtt{L}}, c_{\mathtt{R}})$ and the expected value of the order statistics of the Unif(0, 1) random variables. Given the rational above, we would expect that the optimal pair of hyperparameters $(c^1_{\mathtt{L}}, c^1_{\mathtt{R}})$ at Stage 1 minimizes such a loss.

We present naïve bootstrap method for identifying and merging data for later stages in Algorithm 5 as an alternative to Algorithm 2.

---

**Algorithm 5** Subgroup identification in Stage $t$ $(t > 1)$

---

    **Step 1 (Input):**
1: Input $\{\mathcal{H}_s\}_{s=1}^{t}, \widehat{\tau}_{tj}, \widehat{\mathbb{V}}_{tj}$, and $(c^t_{\mathtt{L}}, c^t_{\mathtt{R}})$ computed from Algorithm 3 or Algorithm 4.
    **Step $b$ (Bootstrap):**
2: **for** $b \leftarrow 1$ to $B$ **do**
3:    Generate totally $\sum_{s=1}^{t} n_s$ resamples randomly with replacement from $\{\mathcal{H}_s\}_{s=1}^{t}$.
4:    Compute $\widehat{\tau}^{\circ}_{tj}$ as in Eq (8) with the bootstrap samples;
5:    Identify the best subgroups $\widehat{\mathcal{T}}_{t1}$ with the bootstrap samples as in Eq (3).
6: **end for**
    **Step $B$ (Output):**
7: Choose $\widehat{\mathcal{T}}_{t1}$ with the highest frequency of occurrence and merge subgroups that belong to $\widehat{\mathcal{T}}_{t1}$.

---

# K   Additional details on synthetic real data study

## K.1   Additional simulation results of our proposed algorithms

In this section, we mainly focus on the comparison of four variations of our proposed design strategy adopted with single or double bootstrap for selecting hyperparameters and naïve or separate bootstrap for data merging. We adopt the same setting mentioned in Section 6.

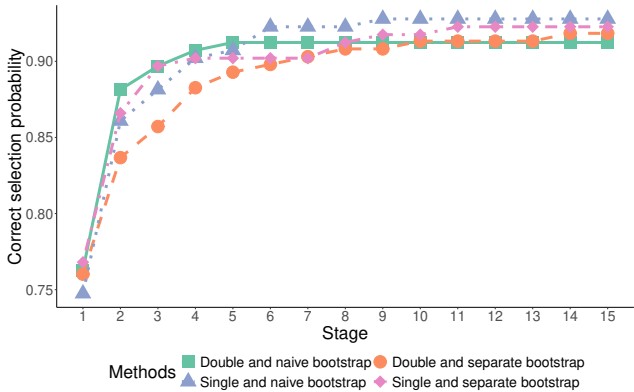

Figure 2: Comparison of the correct selection probability among four variations of our proposed design strategies.

Table 3: Comparison among four variations of our proposed design strategy based on estimated best tie set treatment effect (Est), $95\%$ confidence interval (95% CI), $\sqrt{N}$-scaled bias ($\sqrt{N}$Bias), and standard deviation (SD).

| Method | Est (95% CI) | $\sqrt{N}$Bias | SD |
|---|---|---|---|
| Double and naïve bootstrap | 10.16 (9.14,11.18) | 41.69 | 40.40 |
| Double and separate bootstrap | 10.25 (9.22,11.28) | 34.44 | 40.71 |
| Single and naïve bootstrap | 10.26 (9.22,11.29) | 34.20 | 40.85 |
| Single and separate bootstrap | 10.31 (9.28,11.35) | 29.98 | 40.94 |

First, from the comparison in Figure 2, as the adaptive design goes on, four variations of our proposed design strategy do not present a distinctly different performance, and they are all efficient with correct selection probability gradually rising over 0.9. Our design conducted with a single bootstrap for choosing hyperparameters and naïve bootstrap for identifying and merging tie sets perform a little better than other variations after 15 stages.

Second, from Table 3, we obtain that our proposed strategy conducted with a single bootstrap for selecting hyperparameters and a separate bootstrap for identifying and merging data has the smallest $\sqrt{N}$-scaled bias. Overall, these four variations demonstrate similar efficiency in identifying and merging the best subgroups with almost the same estimated treatment effect and standard deviation.

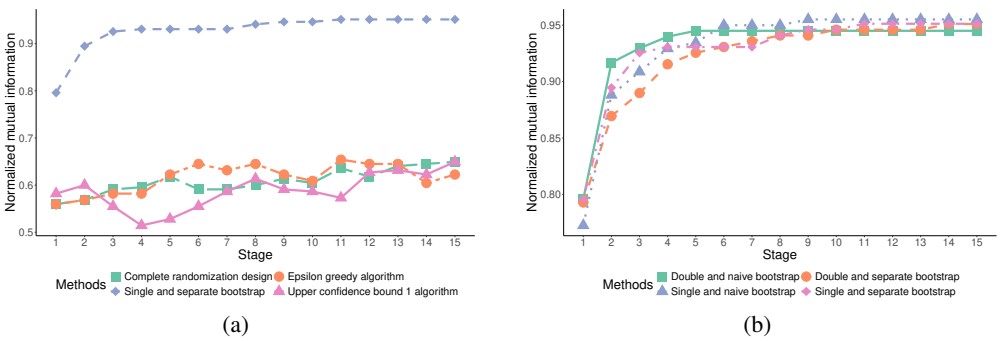

Figure 3: Comparison of the normalized mutual information among three conventional methods and four variations of our proposed design strategies.

We add normalized mutual information as an additional metric to evaluate the similarity of estimated tie set and true tie set of best-performing subgroups. With the results presented in Figure 3 (a) and (b), the normalized mutual information for our proposed design strategy and the three conventional

methods exhibits a similar trend in terms of correct selection probability. This observation further confirms that our proposed design strategy outperforms the conventional methods.

## K.2 Extension: AIPW estimator

When contextual information (or additional covariate information) is considered in our design, the subgroup treatment effects and associated variances in Eq.8 can be replaced by

$$\widehat{\tau}_{t-1,(j)} = \frac{1}{N_{t-1,(j)}} \sum_{s=1}^{t-1} \sum_{i=1}^{n_s} \mathbb{1}_{(X_{is} \in \mathcal{S}_{(j)})} \left\{ \frac{D_{is}}{\widehat{e}_{t-1,(j)}} (Y_{is} - \widehat{\mu}_1(Z_{is})) + \widehat{\mu}_1(Z_{is}) \right\}$$

$$- \frac{1}{N_{t-1,(j)}} \sum_{s=1}^{t-1} \sum_{i=1}^{n_s} \mathbb{1}_{(X_{is} \in \mathcal{S}_{(j)})} \left\{ \frac{1 - D_{is}}{1 - \widehat{e}_{t-1,(j)}} (Y_{is} - \widehat{\mu}_0(Z_{is})) + \widehat{\mu}_0(Z_{is}) \right\}$$

$$\widehat{\mathbb{V}}_{t-1,(j)}(e_j) = \frac{\sum_{s=1}^{t-1} \sum_{i=1}^{n_s} \mathbb{1}_{(X_{is} \in \mathcal{S}_{(j)})} D_{is}(Y_{is} - \widehat{\mu}_1(Z_{is}))^2}{N_{t-1,(j)}(1)} \left( \frac{e_j \cdot N_{t-1,(j)}}{N_{t-1}} \right)^{-1}$$

$$+ \frac{\sum_{s=1}^{t-1} \sum_{i=1}^{n_s} \mathbb{1}_{(X_{is} \in \mathcal{S}_{(j)})} (1 - D_{is})(Y_{is} - \widehat{\mu}_0(Z_{is}))^2}{N_{t-1,(j)}(0)} \left( \frac{(1 - e_j) \cdot N_{t-1,(j)}}{N_{t-1}} \right)^{-1}$$

$$+ \frac{1}{N_{t-1,(j)}} \sum_{s=1}^{t-1} \sum_{i=1}^{n_s} \mathbb{1}_{(X_{is} \in \mathcal{S}_{(j)})} \left( \widehat{\mu}_1(Z_{is}) - \widehat{\mu}_0(Z_{is}) - \widehat{\tau}_{t-1,(j)} \right)^2,$$

where $\widehat{\mu}_d(Z_{is}) = \widehat{\mathbb{E}}[Y_{is}|D_{is} = d, Z_{is} \in \mathcal{S}_j], d \in \{0, 1\}$. We still generate synthetic data that mimic the original PBC dataset with the potential outcome from $Y_i(d)|X_i \in \mathcal{S}_j = \mu_{dj} + 0.5Z_i + \varepsilon_{dj}, \quad Z_i \sim \mathcal{N}(0, 1), \ \varepsilon_{dj} \sim \mathcal{N}(0, \sigma_{dj}^2), j = 1, \ldots, 5$ to investigate the performance of our proposed stratgy with AIPW estimator.

Through simulation studies in Figure 4, we found that our method (whether using IPW or AIPW) has the highest accuracy in identifying the best subgroups. The performance of the causal tree model is comparable to that of the contextual bandit algorithms where treatment effects are estimated with the AIPW estimator, thereby including contextual information. Regarding the causal tree model, we would like to note that the comparison may not be entirely fair for the causal tree model. This is because the causal tree method is a method used to identify subgroups after the data has already been collected, whereas our method focuses on designing data collection mechanisms to accurately identify the best subgroups. Therefore, it can be expected that our approach will have higher accuracy in identifying subgroups.

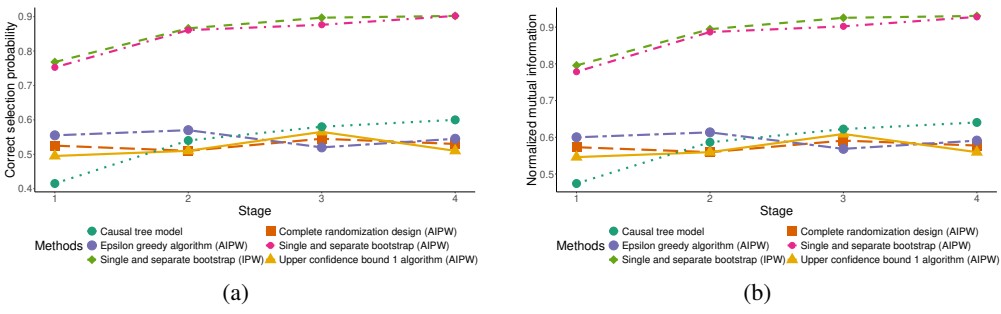

Figure 4: Comparison of the correct selection probability and normalized mutual information among our proposed design with IPW estimator, three conventional algorithms and our proposed design with AIPW estimator and causal tree.

## K.3 Additional synthetic real data study

In this section, we provide an additional synthetic real study in another domain to showcase the performance of our proposed design. We consider a job dataset from the National Supported Work (NSW) program. This program, initiated in the 1970s, aims to provide disadvantaged workers

with work experience. We utilize the dataset from a field experiment referenced in [7] ($n = 455$), which includes 185 workers in the treatment group and 260 workers in the control group. The dataset features a treatment indicator variable, an outcome variable representing participant earnings post-treatment in 1978, and eight baseline variables. These baseline variables include age, years of education, an indicator for high school graduation, indicators for Black and Hispanic ethnicity, marital status, and pre-treatment earnings for the years 1974 and 1975.

We aim to investigate whether the job training program is indeed beneficial for certain groups of workers in for subgroups defined by: (1) education years $\geq 11$ and age $\geq 26$, (2) education years $< 11$ and age $\geq 26$, (3) education years $\geq 11$ and age $< 26$, (4) education years $< 11$ and age $< 26$. It is a suitable dataset because by conducting two-sample t-tests between every two subgroups the pairwise similarity of the distribution of subgroups, Subgroups (1), (2) and (3) are regarded as tie sets of the best subgroups under the significance level of $\alpha = 0.05$. We provide the results in Figure 5, which can validate the performance of our proposed strategy.

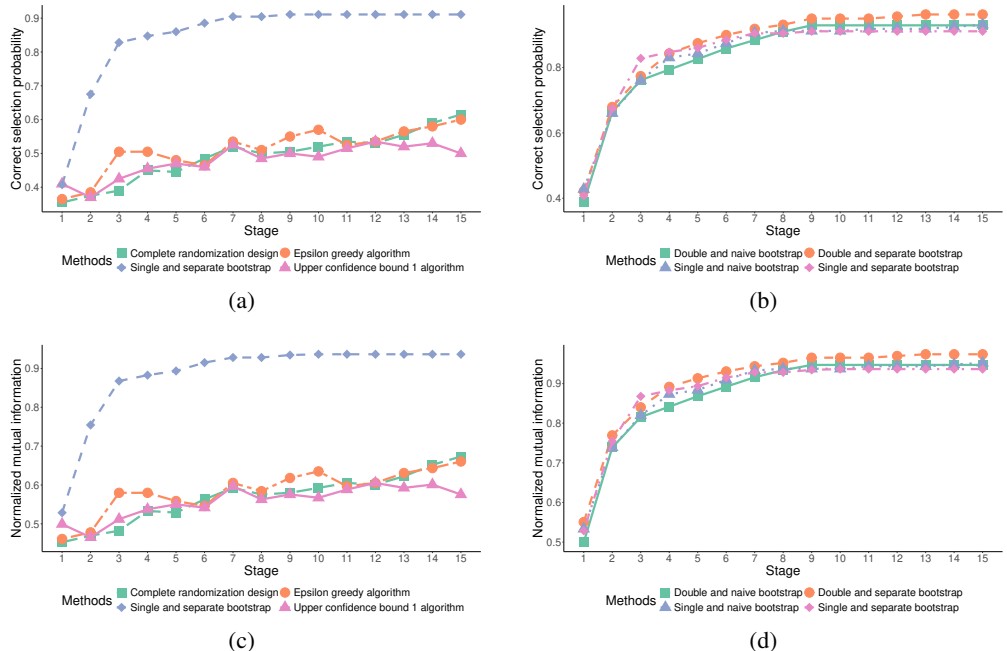

Figure 5: Comparison of the correct selection probability and normalized mutual information among three conventional methods and our proposed design strategy based on job dataset.

