# OpenReview forum: "Dynamic Subgroup Identification in Covariate-adjusted Response-adaptive Randomization Experiments"
_NeurIPS.cc/2024/Conference — NeurIPS 2024 poster_

### Official Review · Reviewer_JR9y · 2024-07-05

**Soundness:** 4
**Presentation:** 3
**Contribution:** 3
**Rating:** 6
**Confidence:** 4

**Summary:**

The paper proposes a dynamic subgroup identification strategy within covariate-adjusted response-adaptive randomization (CARA) for clinical trials. This adaptive method dynamically identifies and adjusts treatment allocation to the best-performing subgroups based on ongoing trial data, and thus extend the traditional fixed-design trials to an online setup. Moreover, the paper also makes theoretical contributions by showing validity, asymptotic normality and efficiency of the estimator for the best subgroup treatment effect.

**Strengths:**

Originality
The paper makes a significant original contribution by introducing a novel dynamic subgroup identification strategy within covariate-adjusted response-adaptive randomization (CARA) for clinical trials. This approach is innovative in its dynamic adjustment of treatment allocation based on real-time data, addressing inefficiencies associated with traditional fixed-design trials. This work also derives new theoretical results justifying the use of their proposed design.

Quality
The research is of high quality, demonstrated through both theoretical and empirical validations. The authors provide rigorous theoretical results, including validity, asymptotic normality and semiparametric efficiency. Additionally, the empirical validation using a synthetic clinical trial on cirrhosis data is well-executed and convincingly demonstrates the effectiveness of the proposed design.

Clarity
The paper is clearly written and well-structured. Complex definitions and algorithms are clearly demonstrated.

Significance
I consider this a significant work in the field of sequential experimental design and subgroup identification. It explores how to best identify and estimate subgroup effects in a dynamic regime, which is underexplored in the literature. What's more, it has broader impact in precision medicine and clinical trials.

**Weaknesses:**

Overall the paper is technically solid, though I notice some issues that can be addressed in the revision. First, the clarity of this paper can be improved as I identified some sentences/equations that are confusing. Please check the question section for details. Second and more importantly, the way that the paper formulates the problem needs some further justification, which is also related to Q2 in the question section.  The paper proposes a design to maximize the probability of correctly identifying the best subgroup, and estimate the best treatment effect; the theory and experiments all focus on this best subgroup identification. However, in personalized medicine which is the motivation for this work, I suppose practitioners care more about whether the treatment is beneficial to a certain group of patients or not, rather than the best treatment effect. In other words, even one can efficiently estimate the best treatment effect using CARA, it does not guarantee the best causal decision rule in personalized medicine. See https://pubsonline.informs.org/doi/10.1287/ijds.2021.0006 for a relevant discussion. Therefore, I suggest the authors discuss this gap between your work and practical considerations; this will help further justify the necessity of best subgroup identification.

**Questions:**

1. Page 3, bottom equation: In the equation at the bottom of this page, should $=$ be $\coloneqq$, i.e., a definition?

2. Page 3, Line 127-128: The goal of the design is to maximize the correct identification probability, based on which the paper proposes CARA and develops the corresponding theory. However, why maximizing the correct identification probability is the ultimate goal? In precision medicine, a more plausible goal is to maximize the overall welfare for the patients, i.e., finding the design that best improves the medical outcome for all patients. How do you compare your design objective to this welfare maximization objective? Can you modify your design for the second objective?

3. Page 4, Eq (1): This equation is confusing. What is the objective function for $\mathbf{e}$? I only saw the constraint set inside the parenthesis.

**Limitations:**

As discussed in the paper, one main limitation is that the method cannot handle delayed outcomes, which can be restrictive in the real-world clinical trials. In my opinion, the authors have adequately addressed the limitations.

---

> ### Author Rebuttal · Authors · 2024-08-06
>
> - Thank you for your insightful questions regarding the design objective and for kindly pointing us to the reference.
>   - We completely understand your concern about the practicality of identifying the best set of subgroups rather than all the benefitted ones. In many clinical settings, treating all the benefitted patients is indeed a natural and ethical goal from the practitioner's standpoint when the number of treatments is unlimited. Different from this setting, our design is tailored to situations where resources are limited, making it preferable to identify only the best set of subgroups. For example, during the onset of COVID-19, when the total number of vaccines was limited, only a subset of the population could be treated. The design objective of identifying the best set of subgroups thus becomes more relevant.
>   -  Furthermore, we would like to point out a subtle difference between our design objective and the design objective of maximizing overall welfare.  Our work considers situations where assigning a treatment is costly, and there is an overall constraint on how many treatments can be deployed.  In Fernandez-Loria and Provost (2022), for example, the goal is to ensure all the benefitted subjects are precisely assigned to the treatment arm while the cost of treatments is negligible. They aim to learn the optimal policy that maps a given covariate to the best arm for that patient profile. In our work, we consider scenarios where implementing the treatment can be costly per sample of a patient (experimentation unit). For example, in clinical settings, randomized experiments are often expensive due to the substantial costs of treatment medications. Consequently, the resource constraints in our problem are equivalent to the number of treatments that can be administered.
>   - We appreciate the reference you provided, which we found very helpful.  We agree that efficiently estimating the best causal effect can sometimes misalign with the best causal decision rule, especially when the best decision rule is to always treat a patient when there is a positive effect. In such cases, accurately estimating the magnitude of the treatment effect becomes less important. Since our primary objective is not to assign all subjects with positive treatment effects to the treatment arm, there could be a misalignment with the best causal decision rule discussed in Fernandez-Loria and Provost (2022). Nevertheless, if we are in a scenario similar to the previously mentioned COVID-19 vaccine example, where the best decision rule is to identify the most benefitted subgroups to prioritize treatment assignments, the accurate causal estimation aligns well with the best causal decision-making. We agree that it is important to be mindful of whether the design objective leads to the best causal decision rule, and the potential misalignment between causal effect estimation and causal decision-making should be carefully discussed. We hope to add this discussion to our revised manuscript.
>   - While our current design is not tailored to welfare maximization, we shall discuss a potential approach to refining our design toward maximizing participant welfare. Consider a two-stage experiment with four candidate subgroups and their population treatment effects follow the order: $\tau\_1=\tau\_2 > \tau\_3 > \tau\_4$, where $\tau\_1 = \tau\_2 > \tau\_3 > 0 \geq  \tau\_4$. At the end of Stage 1, our current procedure will be able to correctly identify subgroups 1 and 2 as the best set of subgroups with a high probability. If we were to maximize the patient's welfare, we would add an additional ``early stopping" step. That is, besides identifying the best set, we will also identify subgroups that exhibit significantly adverse treatment effects. To protect the patient's welfare, we will stop Subgroup 4 from enrollment in the next stage. In Stage 2, we not only avoid impairing the welfare of Subgroup 4 but also maximize the resources (treatments) available to the rest of the benefitted subgroups.
> - Thank you for your careful reading of our manuscript. We shall clarify our optimization problem formulation as follows. Our original optimization problem is formulated as $\max_{ \mathbf{e}} \min\_{2\leq j\leq m^\ast\_{t}} \frac{( \hat{\tau}\_{t-1,(j)} -  \hat{\tau}\_{t-1,(1)})^2}{2\big(\hat{\mathbb{V}}\_{t-1,(1)}(e\_{1}) + \hat{\mathbb{V}}\_{t-1,(j)}(e\_{j})\big)}, \text{s.t.}\ \sum\_{l=1}^{m^\ast\_{t}} \hat{p}\_{tl} e\_l \leq c\_1,    \  c_2\leq e\_l \leq 1-c\_2,\ l=1, \ldots,m^\ast\_{t}$. The set of constraints includes the resource constraint: $\sum\_{l=1}^{m^\ast\_{t}} \hat{p}\_{tl}\leq c\_l$, and the feasibility constraint: $c_2\leq e\_l \leq 1-c\_2,\ l=1, \ldots,m^\ast\_{t}$.  Because the original objective function takes the minimum of $m^*-1$ rate functions, the original optimization problem is nonlinear. We instead work with its equivalent epigraph representation: $\max\_{ \mathbf{e}} z$, s.t. $ \min\_{2\leq j\leq m^\ast\_{t}}\frac{( \hat{\tau}\_{t-1,(j)} -  \hat{\tau}\_{t-1,(1)})^2}{2\big(\hat{\mathbb{V}}\_{t-1,(1)}(e\_{1}) + \hat{\mathbb{V}}\_{t-1,(j)}(e\_{j})\big)} -z \geq 0,  \ \sum\_{l=1}^{m^\ast\_{t}} \hat{p}\_{tl} e\_l \leq c\_1,    \  c_2\leq e\_l \leq 1-c\_2,\ l=1, \ldots,m^\ast\_{t}$, which is Eq (1) in our current submission. The original formulation of the objective function was omitted due to space limits. We will clarify the objective function in our revision.
>
> Per your suggestions, we plan to make the following updates to our manuscript:
> - We will add a discussion on the potential gap between our design and the best causal decision rule in precision medicine scenarios in our revised manuscript.
> - We will revise sentences and notations for clarity.
> - We will provide more illustrations of Eq (1).
>
> Reference:
> - Fernandez-Loria, C. and Provost, F. (2022). Causal decision making and causal effect estimation are not the same... and why it matters. INFORMS Journal on Data Science, 1(1):4–16.

---

> > ### Comment · Reviewer_JR9y · 2024-08-09
> >
> > Thanks for the clarification, especially on the distinctions between your work and welfare maximization objective. I have no further concerns.

---

### Official Review · Reviewer_Sgje · 2024-07-06

**Soundness:** 3
**Presentation:** 3
**Contribution:** 3
**Rating:** 7
**Confidence:** 2

**Summary:**

This paper introduces a new dynamic treatment assignment for clinical trial to target treatment to the group most likely to benefit from it.

**Strengths:**

The paper studies a critical problem of clinical trial design. It is clear and provides both theoretical justification and synthetic validation, demonstrating the utility of the proposed method.

**Weaknesses:**

While the paper tackles a critical problem, the problem is studied in depth in biostatistics. While I am unfamiliar with the literature on this topic, I am surprised there is none that could be considered for comparison. A more in-depth analysis of the literature on this topic should be presented in the Appendix to justify the choice of compared methods.

**Questions:**

Following the previous point, I would recommend a more in-depth review of the literature to convince the reader that this problem does not have alternative solution in the literature.

**Limitations:**

The paper discusses the critical limitation of the proposed approach's assumption of instantaneous access to the outcome following treatment. It would be interesting to discuss this assumption in the context of the existing literature. Is it a common assumption? If not, it would be important to justify further.

---

> ### Author Rebuttal · Authors · 2024-08-06
>
> Thank you for your valuable questions and suggestions!
>
> A main limitation of our proposed approach is the assumption that outcomes are observed instantaneously following treatment. This assumption, prevalent in adaptive experiments such as Hu et al. (2015) and Zhu and Zhu (2023), simplifies the modeling process and allows for quick adjustments based on the latest data. While this assumption is common, it may not always reflect real-world scenarios where response delays. To provide a more comprehensive understanding, we review the literature addressing delayed responses.
>
> - The importance of incorporating delayed responses in adaptive experiments is well-recognized in the literature. Rosenberger et al. (2012) discuss the effects of delayed responses on response-adaptive randomization. Early work by Wei and Durham (1978) introduces the randomized play-the-winner rule, which updates the contents of an urn only upon receiving patient responses, thus naturally accommodating delayed responses. This approach offers inherent flexibility in managing delays and represents an improvement over more rigid methods. Bai et al. (2002) and Hu and Zhang (2004) establish asymptotic normality results under urn models with delayed responses, explicitly concerning the asymptotic normality of the fraction of patients assigned to each treatment arm. Their findings show that limiting distributions remain unaffected by delayed responses under reasonable conditions, providing a solid theoretical basis for handling such delays. Zhang et al. (2007) extend this work by introducing a generalized drop-the-loser urn model, demonstrating that asymptotic properties are preserved despite response delays. This generalized model offers broader applicability and enhanced flexibility for managing various delay mechanisms.
> - Regarding the doubly-adaptive biased coin design (DBCD), Zhang and Rosenberger (2006) have shown through simulations that moderate delays in responses have minimal impact on the power and skewness of the DBCD. Their results suggest that the design remains robust even in the presence of delays, although the effects of more severe delays are less clear. Hu et al. (2008) study the asymptotic properties of the DBCD with delayed responses, showing that these properties remain unaffected by such delays. They also provide strong consistency results for the constructed variance estimator that incorporates delayed
> responses, enhancing the design's reliability in practical scenarios.
> - In addition to response-adaptive randomization designs, some group-sequential designs address the challenges brought by delayed responses. Hampson (2013) proposes to incorporate short-term endpoints to enhance the efficiency of group-sequential tests when long-term responses are delayed. This method effectively balances immediate data needs with constraints imposed by delayed outcomes. Schuurhuis et al. (2024) expand this framework by suggesting the integration of pipeline data into group-sequential designs, enabling the trial to restart after a temporary halt. This integration provides added flexibility and robustness in managing trials with delayed responses.
>
> - Overall, handling delayed responses requires challenging adjustments to our design. First, our objective function, based on the semiparametric efficiency bound of the subgroup treatment effect, must be revised. The difficulty stems from the function's current assumption of immediate responses, and modifying it to account for delays significantly complicates the task of ensuring accuracy.
> Second, our estimators need updating. Delayed responses impact both the treatment effect and variance estimators, necessitating an additional step to address these delays. While this step is crucial for maintaining precision and reliability, it also introduces complexities that must be managed to ensure a robust estimation process.
>
> Per your suggestions, we plan to make the following updates to our manuscript:
> - Present a more in-depth analysis of the literature in our revised manuscript.
> - Discuss the critical limitation in the context of the existing literature in our revised manuscript.
>
> Reference:
> - Hu, J., Zhu, H., and Hu, F. (2015). A unified family of covariate-adjusted response-adaptive designs based on efficiency and ethics. Journal of the American Statistical Association, 110(509):357–367.
> - Zhu, H. and Zhu, H. (2023). Covariate-adjusted response-adaptive designs based on semiparametric approaches. Biometrics.
> - Rosenberger, W. F., Sverdlov, O., and Hu, F. (2012). Adaptive randomization for clinical trials. Journal of Biopharmaceutical Statistics, 22(4):719–736.
> - Wei, L. and Durham, S. (1978). The randomized play-the-winner rule in medical trials. Journal of the American Statistical Association, 73(364):840–843.
> - Bai, Z., Hu, F., and Rosenberger, W. F. (2002). Asymptotic properties of adaptive designs for clinical trials with delayed response. The Annals of Statistics, 30(1):122–139.
> - Hu, F. and Zhang, L.-X. (2004). Asymptotic normality of urn models for clinical trials with delayed response. Bernoulli, 10(3):447–463.
> - Zhang, L.-X., Chan, W. S., Cheung, S. H., and Hu, F. (2007). A generalized drop-the-loser urn for clinical trials with delayed responses. Statistica Sinica, 17(1):387–409.
> - Zhang, L. and Rosenberger, W. F. (2006). Response-adaptive randomization for clinical trials with continuous outcomes. Biometrics, 62(2):562–569.
> - Hu, F., Zhang, L.-X., Cheung, S. H., and Chan, W. S. (2008). Doubly adaptive biased coin designs with delayed responses. Canadian Journal of Statistics, 36(4):541–559.
> - Hampson, L. V. and Jennison, C. (2013). Group sequential tests for delayed responses (with discussion). Journal of the Royal Statistical Society Series B: Statistical Methodology, 75(1):3–54
> - Schuurhuis, S., Konietschke, F., and Kunz, C. U. (2024). A two-stage group-sequential design for delayed treatment responses with the possibility of trial restart. Statistics in Medicine.

---

> > ### Comment · Reviewer_Sgje · 2024-08-12
> > **Maintain score**
> >
> > Thank you for adding the literature review, I am maintaining my score

---

### Official Review · Reviewer_Rfeb · 2024-07-11

**Soundness:** 2
**Presentation:** 1
**Contribution:** 2
**Rating:** 4
**Confidence:** 5

**Summary:**

This paper studies an interesting problem in clinical trials to identify patient subgroups with the most beneficial responses to the treatment, which is essential for clinicians to create personalized treatment plans for their patients. However, most existing strategies for the design of clinical trials rely on domain knowledge or past experience of expert clinicians and stick to several pre-defined patient subgroups throughout the trial, which discards the valuable information collected from different trial stages. Some adaptive experimental strategies are developed to identify the best performing patient subgroup based on the trial outcomes; but their negligence of other subgroups where the treatment could be equally effective usually causes inefficient utilization of the experimental efforts. To tackle these challenges, the authors propose a dynamic subgroup identification approach to enable the construction of more effective experimental strategies for practical clinical trials. The authors claim three major contributions in this study: 1) their approach allows the dynamic identification for best patient subgroups based on experimental data collected during the trial process; 2) the authors develop new algorithms to effectively merge patient subgroups with similar (highest) responses to the treatment and provide theoretical results to support their analyses; and 3) the proposed method is validated using a synthetic dataset constructed from a clinical trial on cirrhosis.

**Strengths:**

## Originality
The method presented in this paper looks to be novel. The authors have provided substantial theoretical analysis to evident the originality and validity of their approach.

## Quality
Most parts of this paper are well-written and properly organized. The experimental results are reported with relevant statistics and are clearly evaluated and discussed with texts and visualizations.

## Clarity
The general clarity of this paper is fair. Experimental results in this paper are sufficiently discussed.

## Significance
The problem of patient subgroup identification is essential in clinical trials for the selection of target patient cohorts. For better utilization of the resources in a trial, dynamic identification of best performing patient subgroups and adaptive treatment assignments are imperative. The approach proposed in this paper enables effective patient subgroup identification using patient characteristics and treatment responses collected during different trial stages and allows adaptive optimization of the treatment assignment strategy.

**Weaknesses:**

## Related Works

There are two short paragraphs in the Introduction discussing the literature related to this study. However, to distinguish the experimental design approach in this paper from other research and highlight the contributions of this study, a more comprehensive comparison with related works is needed. It seems that there are many other studies, with similar focuses on subgroup identification in clinical trials, not sufficiently discussed in this paper. For instance:

- Adaptive identification and assessment of patient subgroups [1]
- Identification of patient subgroups with similar clinical characteristics (covariates) [2]
- Clustering of patient subgroups with different levels of benefits in clinical trials [3]

The authors are encouraged to provide discussions and comparisons with related works on patient subgroup identification to better demonstrate the novelty and advantage of this study.

Further, although the authors emphasize that their method focuses on the setting of covariate-adjusted response-adaptive (CARA) experiments which is an under-explored area, the reviewer cannot find discussions on the contribution or benefit of including patient covariates ($X$) in the design of clinical trials. The covariates-based patient subgroup identification has already been studied in the machine learning literature, e.g., [2,3].

References:

[1] Guo, Wentian, Yuan Ji, and Daniel VT Catenacci. "A subgroup cluster‐based Bayesian adaptive design for precision medicine." Biometrics 73.2 (2017): 367-377.

[2] Lee, Beom S., et al. "A clustering method to identify who benefits most from the treatment group in clinical trials." Health Psychology and Behavioral Medicine: an Open Access Journal 2.1 (2014): 723-734.

[3] Xu, Jie, et al. "Machine learning enabled subgroup analysis with real-world data to inform clinical trial eligibility criteria design." Scientific Reports 13.1 (2023): 613.





## Clarity

### The role of patient covariates

As mentioned earlier, the role of patient covariates is not sufficiently discussed in the proposed experimental design approach. The only place related to the covariates seems to be line 7 of Algorithm 2 where previous trial results are randomly resampled. It is unclear how the experimental design is adjusted based on patient covariates. Since this paper focuses on the CARA setting and considers both treatment responses and patient covariates in subgroup identification, it is important for the authors to elaborate the contribution of patient covariates in the proposed algorithms and how is this study different from conventional research on response-adaptive randomization (RAR) settings.

### Notations

There are many symbols introduced in the analysis of this paper without any explanation.
For instance, the variable $\hat{p}\_{tl}$ in Eq. 1 has never been explained.
The symbol $\mathcal{B}_{t,b}$ in Eq. 6 comes from nowhere.
In the meantime, there are so many similar symbols used in the discussion, and it is very difficult for the reviewer to tell their difference. The authors are encouraged to ensure the consistency in their notation. If possible, a notation table could help a lot to improve the clarity of this paper.

### Insufficient explanations

Some results or derivations in this paper are introduced without proper explanation. For instance, although the authors provide citations to the large deviation theory in LN 129, the equivalence between the correct identification probability and the optimization objective remains obscure. Similarly, it is unclear why the optimal treatment allocation can be derived from Eq. 1. Additionally, the hard-coded exponent 0.05 for $\Delta$ in LN 162 appears without any explanation.

## Correctness

The correctness of some proofs seems to be questionable. For instance, in the proof of Theorem 1, it is unclear why $\tau_{t,l} = \tau_{l}$. The inequality in LN 450 cannot be directly established according to the analysis in LN 451 – 453. Similarly, the final inequality in LN 457 is non-trivial and the authors should provide analysis to prove its correctness.

## Evaluation

### Dataset

The proposed experimental design approach is only evaluated on a synthetic dataset. To validate the general applicability and performance of this method, benchmark results on more datasets are necessary.

### Baselines

The authors have discussed many relevant studies in the Introduction. However, only three baselines are considered in the experiment. For a more comprehensive comparison, the authors are encouraged to include more baselines from related works to demonstrate the advantage of their method. Particularly, the reviewer is interested in the performance of contextual bandit algorithms and causal tree models.

### Metrics

Note that the experimental results are obtained on a synthetic dataset where the ground truth subgroup labels are available. The authors are encouraged to include additional metrics on clustering accuracy, e.g., purity score, normalized mutual information, etc., in the benchmark.

**Questions:**

In summary, I have the following concerns about this paper.
1. The related works are not sufficiently discussed. The difference between this paper and relevant studies should be clearly illustrated.
2. It is unclear how the proposed method is different from conventional methods considering the RAR settings. There is no discussion on the contribution or benefit of including patient covariates ($X$) in the design of clinical trials.
3. Many symbols are introduced in the analysis of this paper without proper explanation.
4. The derivation of some key results is not sufficiently explained. For instance, the optimization objective below LN 129 and the optimal treatment allocation in Eq. 1.
5. The correctness of some proofs seems to be questionable. Specifically, the proof for Theorem 1. Please see the weakness section above for details.
6. The proposed method needs to be tested on more datasets, and the authors are encouraged to include more performance metrics in the benchmark.
7. There should be more baselines, e.g., causal tree and contextual bandit, in the benchmark to highlight the advantage of the proposed method.
8. How does the number of stages $T$ in a trial affect the convergence of the proposed method? What if there is only one stage? What if $T$ is small (Below 4)?

**Limitations:**

The limitations of this study are briefly discussed at the end of this paper. The authors identify the potential mismatch between their assumption on immediate observation of treatment responses and the delays commonly observed in real-world clinical trials. However, the potential negative societal impact of the clinical trial design approach in this paper is not sufficiently discussed. In addition, the authors are encouraged to discuss whether their approach can be generalized to deal with scenarios with multiple treatment options.

---

> ### Author Rebuttal · Authors · 2024-08-06
>
> 1. Thank you for pointing us to the references. While both Guo et al. (2017) and ours are in an adaptive experiment setting, there are two major differences. (i)Their design is Bayesian, relying on prior specification, while ours is frequentist and model-free. (ii)They do not discuss theoretical properties for the identified subgroups, while we provide theoretical results justifying the reliability of our approach. Next, there are two lines of literature for subgroup identification: (1)post-hoc analyses using previously collected data, and (2)adaptive data collection through randomized experiments. Lee (2014) and Xu (2023) align with the first one, whereas our method focuses on the second. Lee et al. (2014) use clustering techniques based on data from prior randomized controlled trials, and Xu et al. (2023) identify subgroups from existing observational data using machine learning, both differing from our adaptive experiment setting.
>
> 2. In classical CARA design (so does our design), patient covariates are used to define subgroups. Concretely, denote the covariate as $X_{it}$ and assume the covariate space $\mathcal{X}$ is partitioned into $m$ regions: $\mathcal{S}_{j=1}^m$ (lines 98-100).  CARA designs differ from RAR as RAR does not consider covariates at all, optimizing treatment allocation solely based on past treatment assignments and outcomes. This oversight can lead to less effective treatment strategies by ignoring valuable patient-specific information that could significantly influence treatment responses. We hope to note that in response to your Q7, we refined our approach using the augmented Inverse Probability Weighting estimator, allowing us to incorporate other covariates that are not used to define subgroups, enabling us to adjust treatment allocation accordingly.
>
> 3.We apologize for missing the definition of $\hat{p}_{tl}$. It is defined as $\hat{p}\_{tl}=\frac{\sum\_{s=1}^{t}\sum\_{i=1}^{n\_s}\mathbb{1}\_{(X\_{is}\in\mathcal{S}\_{(l)})}}{\sum\_{s=1}^{t}n_s}$. ${B}\_{t,b}$ is defined in Eq.6 in preparation for the calculation of the loss function in Eq.7. Additionally, $\Delta=\min(1,R\cdot(\frac{\sqrt{\sum\_{j=1}^{m}n\_{tj}\hat{\mathbb{V}}\_{tj}/m}}{\sqrt{\sum\_{j=1}^{m}\hat{\mathbb{V}}\_{tj}/n}})^{2*\gamma})\approx \min(1,R\cdot n^{\gamma})$, where $\gamma$ is a small tuning parameter to ensure $\Delta<1$. We choose $\gamma=0.05$ and show that our procedure is not sensitive to the tuning parameter(Table 1 in the attached pdf).
>
> 4. Due to the space limit in our first submission, we neglected technical details to justify Eq.1. In fact, the rate function in the optimization problem's objective is a monotone transformation of the correct identification probability in an asymptotic sense. Below is a brief derivation due to the space limit: $\lim\_{N\rightarrow\infty}\frac{1}{N}\log(1-\mathbb{P}( \hat{\tau}\_{\mathcal{T}\_1}\geq\max\_{j \notin\mathcal{T}\_1}\hat{\tau}\_j))=-\min\_{j\notin \mathcal{T}\_1}G(\mathcal{S}\_1,\mathcal{S}\_j;e\_1,e\_j)$. We are happy to provide more details if you raise additional concerns.
>
> 5. After inspecting our proof, we believe that our proof in the three places you pointed to us is correct. $\tau\_{t,l}=\tau\_l$ naturally holds by Assumption 1. This is because potential outcomes are independently identically distributed for $i=1,\ldots,n_t$, $t=1,\ldots,T$, implying $\tau\_{t,l}=\tau\_l$ in the proof of Theorem 1. This i.i.d assumption is commonly assumed in adaptive design literature.  With the analysis in LN 451-453, we have $\lim\_{N\rightarrow\infty}\mathbb{P}(N^{\delta}|\tau\_{j}-\tau\_{\check{1}}|<C)=0$ and thus $\lim\_{N\rightarrow\infty}\mathbb{P}( N^{\delta}|\tau\_{j}-\tau\_{\check{1}}|+C< 2C)=0$. By Lemma 2 in the Appendix, we obtain $\lim\_{N\rightarrow\infty}\mathbb{P}(N^{\delta}|\hat{\tau}\_{tj}^*-\tau\_{j}| +N^{\delta}|\hat{\tau}\_{t,\check{1}}^*-\tau\_{\check{1}}|\geq 2C)=0$. Then we reach the conclusion in Eq.13. Similarly, with the analysis in LN 458-461, we have $\lim\_{N\rightarrow\infty}\mathbb{P}( N^{\frac{1}{2}}|\tau\_{j}-\tau\_{\check{1}}|<C)=1$. We also derive
>
> $\lim\_{N\rightarrow\infty}\mathbb{P}(N^{\delta}|\hat{\tau}\_{tj}^*-\tau\_{j}|<C)=1$ and
> $\lim\_{N\rightarrow\infty}\mathbb{P}(N^{\delta}|\hat{\tau }\_{t,\check{1}}^*-\tau\_{\check{1}}|<C)=1$. With $\delta<\frac{1}{2}$, we establish the result in LN 455.
>
> 6. We have added an additional case study using the National Supported Work program data(Figure 1 in the attached pdf).
>
> 7. For the comparison with various contextual MAB algorithms, we now extend our proposed design to an augmented inverse propensity score weighting (AIPW) estimator incorporating contextual information (Figure 2 in the attached PDF). For the causal tree, the comparison may not be entirely fair since the causal tree identifies subgroups after data collection, whereas our method designs the data collection mechanism to identify the best subgroups accurately. Thus, our approach is expected to have higher accuracy in identifying subgroups. We have provided a comparison (Figure 2 in the attached PDF). The performance of the causal tree model is similar to contextual bandit algorithms, and our proposed algorithm has the highest probability of identifying subgroups.
>
> 8. The purity score is similar to correct selection probability, as both measure accuracy: purity score assesses how well clusters contain a single class, while correct selection probability evaluates identifying the best subgroups. Both range from 0 to 1, with higher values indicating better performance. Figure 1(c) (d) in the attached PDF compares the normalized mutual information for our proposed design and three competing methods.
>
> Per your suggestions, we will update our manuscript as follows:
> - Add a more thorough literature review on posthoc subgroup analysis
> - Add a notation table
> - Add additional simulation results, including contextual bandit, AIPW estimator, and normalized mutual information as an additional metric

---

> > ### Comment · Reviewer_Rfeb · 2024-08-12
> >
> > Thank you for your response. I appreciate the authors' clarification on related works and details in their proof.
> > However, regarding the new results presented in the rebuttal, I still have the following concerns.
> >
> > #### The usage of patient covariates
> > Thanks for the clarification on the differences between the proposed method and RAR and CARA approaches.
> > It seems that the trial strategy proposed in this paper relies on some conventional covariate-adjusted strategies to generate the initial subgroup division.
> > However, its robustness with respect to the subgroup initialization is not properly evaluated.
> > What if the initial subgroups are not correctly aligned to the distribution of treatment responses in a population (which could be common in real-world applications)?
> >
> > #### Tuning parameter $\\gamma$
> > The authors mention that the number $\\gamma=0.05$ is a small tuning parameter to ensure that the bootstrap factor $\\Delta<1$ in Eq. (3) and have provided ablation study about $\\gamma$. However, this tuning parameter seems to be useless according to Table 1 provided in the authors' rebuttal. As affirmed by the authors, their proposed procedure is insensitive to $\\gamma$. When $\\gamma=1$, $\\Delta$ could always be equal to 1, which completely discards the proposed bootstrap strategy. This raise further concerns on the novelty of this paper.
> >
> > #### Extension with augmented IPW.
> > Comparing Fig. 1 in the main manuscript and Fig. 2(a) in the authors' rebuttal, the inclusion of IPW estimator leads to no obvious improvement in performance.
> > Why is this happening? Contextual information shall allow more precise treatment allocation during the trial and contributes to faster convergence (e.g., high correct selection probability with fewer stages). Therefore, the improvement mentioned in the authors rebuttal doesn't seem to be effective.
> > Further, this may suggest that the synthetic dataset used in the experiment is inappropriate for serious performance evaluation.

---

> > > ### Author Response · Authors · 2024-08-13
> > >
> > > Thank you for raising additional concerns. Due to space constraints, we hope to break down our response into two comments. The first comment below addresses your first and third concerns.
> > >
> > > **Usage of covariates**: To ensure our proposed design is practically relevant, our design does not incorporate any data-driven methods for identifying initial subgroup divisions after the trial has started. This is because regulatory agencies often strongly encourage that subgroups be pre-specified to enhance the interpretability of trial results and prevent data mining during the planning phase of a clinical trial; see some sample RCT designs with pre-fixed subgroups in Murray et al.(2018), Thall et al.,(2003), and Hu et al.,(2015).In accordance with this regulation, our design aims to dynamically merge subgroups if certain subgroups show homogeneous effects (note that merging does not create new divisions of subgroups) and sequentially adjusts treatment assignment probabilities within those merged subgroups to identify the best one efficiently. We do see that adding an initialization stage can be helpful when the initial subgroups are not informative of the treatment effect heterogeneity. As a future direction, we plan to explore the possibility of using tree-based methods to identify subgroups and sequentially merge them.
> > >
> > > **Extension with augmented IPW**: Indeed, it may seem counterintuitive that including additional covariates does not significantly improve the empirical performance. This is because our proposed method employs the IPW estimator with estimated propensity scores, which, as justified by Hirano et al. (2003), already attains the semiparametric efficiency bound. Thus, in the supplementary simulation study, adjusting for additional covariates with AIPW does not further improve the empirical performance of the treatment effect estimator, and therefore, the empirical performance of our design remains unchanged. We hope this addresses your concern! Thank you very much for carefully going through our new simulation results.
> > >
> > > Reference:
> > > - Thall, P. F., Wathen, J. K., Bekele, B. N., Champlin, R. E., Baker, L. H., and Benjamin, R. S.(2003). Hierarchical bayesian approaches to phase ii trials in diseases with multiple subtypes. Statistics in medicine, 22(5):763–780.
> > > - Murray, T. A., Yuan, Y., Thall, P. F., Elizondo, J. H., and Hofstetter, W. L. (2018). A utility-based design for randomized comparative trials with ordinal outcomes and prognostic subgroups.Biometrics, 74(3):1095–1103
> > > - Hu, J., Zhu, H., and Hu, F. (2015). A unified family of covariate-adjusted response-adaptive designs
> > > based on efficiency and ethics. Journal of the American Statistical Association, 110(509):357–367
> > > - Hirano, K., Imbens, G. W., and Ridder, G. (2003). Efficient estimation of average treatment effects using the estimated propensity score. Econometrica, 71(4):1161–1189.

---

> ### Author Response · Authors · 2024-08-13
>
> The second comment below addresses your second concern:
>
> **Tuning parameter**: We are sorry that the previous presentation of our procedure might have caused some confusion. In the following, we would like to clarify that our bootstrap procedure is not intended to select $\gamma$, and both $\gamma$ and $\Delta$ are only adopted to select the hyperparameter pair $(c\_\texttt{L},c\_\texttt{R})$, which determines the neighborhood for merging subgroups.
>
> First, we will replace lines 159-170 with the following in our revision to clarify our procedure:
>
> - **Dynamic identification of the best subgroups (Algorithm 2)**:The dynamic subgroup identification algorithm involves a resampling step at each stage. Specifically, at each stage $t$, we generate bootstrap samples  $\hat{\mathbf{\tau }}\_t^{\circ}$ from a Gaussian distribution centering around $\hat{\mathbf{\tau }}\_t$, which is estimated using the data collected up to stage $t$ using Eq (8). We then identify the best subgroups at Stage $t$ using
> \begin{align*}
> \hat{\mathcal{T}}\_{t1} = \\{k: w\_{k,(1) }^{\circ} = 1,k = 1,\ldots,m\\}, \ (5)
> \end{align*} where $w\_{k,(1)}^{\circ}=\mathbb{1}\\{-c^t\_{\texttt{L}}\cdot N\_t^{-\delta}\cdot \hat{\mathbb{V}}\_{t,( 1)}^{\delta}\leqslant (\hat{\tau}\_{tk}^{\circ}-\hat{\tau}\_{t,(1) }^{\circ})\leqslant c^t\_{\texttt{R}}\cdot N\_t^{-\delta}\cdot \hat{\mathbb{V}}\_{t,(1)}^{\delta }\\}$. $N_t = \sum\_{s=1}^t n\_s$ and $\delta = 0.25$. Note that the above formulation relies on a pair of hyperparameters $(c^t\_{\texttt{L}},c^t\_{\texttt{R}})$, which are selected data-adaptively. In what follows, we shall illustrate the algorithm for selecting hyperparameters.
>
> - **Hyperparameter selection (Algorithm 3)**: In line 7 of Algorithm 3, we adopt a bootstrap method and propose several alternative bootstrap methods in Algorithm 3(line 7) in Appendix (Section H).Algorithm 3 involves a resampling step that generates bootstrap samples $\hat{\mathbf{\tau}}\_t^*$  from a Gaussian distribution centering around $\mathbf{\tau}\_t^*$ at Stage 1. In line 2, we compute $\mathbf{\tau}\_t^*=(\tau\_{t1}^*,\ldots,\tau\_{tm\_t^*}^{* })^{\prime}$ as \begin{align*} \tau\_{tj}^*=\Delta\_t \cdot \frac{\sum\_{j=1}^{m\_t^*}\hat\tau\_{tj}}{m\_t^*}+(1-\Delta\_t) \cdot \hat\tau\_{tj}, \ j = 1,\ldots,m, \ (3) \end{align*} where $\Delta\_t =\min\\{0.99,\frac{\sum\_{j=1}^{m\_t^*}\hat{\mathbb{V}}\_{tj}}{ N\_t\sum\_{j=1}^{m\_t^*}(\hat{\tau}\_{tj}-\overline{\hat{\tau}}\_t)^{2}}\times N\_t^{\gamma}\\}$ and $\gamma \in (0,0.2)$. We choose $\gamma = 0.05$ in our simulation studies, and our procedure is shown not sensitive to $\gamma <1$.  In line 8, we compute  $\hat{\mathbf{\tau}}\_t^*=(\hat{\tau}\_{t1}^*,\ldots,\hat{\tau}\_{tm\_t^*}^*)^{\prime}$ at Stage $t$ for $t>1$ as
> \begin{align*}
> \hat{\tau}\_{tj}^*=\Delta\_t \cdot \frac{\sum\_{j=1}^{m\_t^*}\hat{\tau}\_{tj}^{\circ}}{m\_t^*}+(1-\Delta\_t) \cdot \hat{\tau }\_{tj}^{\circ},  \ (4)
> \end{align*} where $\hat{\tau}\_{tj}^{\circ}$ is computed with the bootstrap samples as in Eq (8).
>
> Through this revision, we hope to clarify that
> - The final tie set is selected using Eq (5), not depending on $\Delta$ and $\gamma$.
> - You are absolutely correct that $\Delta$ cannot be 1; we will revise our algorithm to put a lower bound of 0.99. Additionally, there was a typo in Table 1; the magnitude of $\gamma$ we have tested is actually from $0.00$ to $0.10$. We hope to provide additional simulation results below. Due to time limit, we are only able to provide results for reduced sample size with $n_t = 400$, $T=4$ and reduced $B=400$ on the correct selection probability, Monte Carlo bias, and variance for a various choice of $\gamma$:
>   -  $\gamma = 0.05$, CSP: $0.68, 0.81, 0.85, 0.86$; $\sqrt{N}$Bias: $35.20$; SD: $44.71$;
>   - $\gamma = 0.10$, CSP: $0.68, 0.81, 0.85, 0.86$; $\sqrt{N}$Bias: $35.20$; SD: $44.71$;
>   -  $\gamma = 0.15$,CSP: $0.67, 0.81, 0.86, 0.86$; $\sqrt{N}$Bias: $35.21$; SD: $44.72$;
>   - $\gamma = 0.20$, CSP: $0.68, 0.81, 0.86, 0.86$; $\sqrt{N}$Bias: $34.42$; SD: $44.75$.
>
> The performance of our method in this response may not be as strong as the results in the submitted manuscript due to time constraints in addressing your concerns. The results still demonstrate that our method is not sensitive to the choice of $\gamma$.
>
> We hope this revision of our manuscript will address your concerns!

---

> > ### Comment · Reviewer_Rfeb · 2024-08-14
> >
> > I appreciate the authors' additional rebuttal which has addressed my concerns on subgroup initialization and the contribution of AIPW.
> >
> > #### More effective baselines
> > According to the authors' response, a more reasonable baseline should be contextual bandits with AIPW estimation as feature maps for treatment assignment (and using clustering algorithms like K-means or agglomerative clustering to merge subgroups with similar treatment responses as desired in this paper). The current baselines seem to be quite weak.
> >
> > #### Tuning parameter
> > I am not fully convinced by the authors' response.
> > - **Tie set selection**: The tie set selection in Eq. (5) is dependent of $\\hat{\\tau}^{o}_{tk}$ which is computed with the bootstrap samples as clarified by the authors. This implies that the tie set selection is in fact affected by $\\Delta$ and $\\gamma$.
> > -  **Upper bound of $\\Delta$**: Setting the upper bound of $\\Delta$ to 0.99 is kind of arbitrary and lacks rigor.
> > - **Typo in new results**:  I appreciate the new results by the authors. However, it is difficult to verify whether the range of parameter $\\gamma$ reported in Table 1 of the rebuttal is a typo.
> >
> > Therefore, I can only increase my rating of this paper from 3 to 4.

---

> > > ### Author Response · Authors · 2024-08-14
> > >
> > > Thank you for your reply and recognition. We regret that there are still concerns regarding our paper.
> > >
> > > **More Effective Baselines**
> > >
> > > Thank you for suggesting an additional baseline. We will include a comparison with this baseline in the revised manuscript.
> > >
> > > **Tuning Parameter**
> > >
> > > - **Tie Set Selection:** Our hyperparameter selection process is independent of the dynamic identification of the best subgroups. Specifically, there are two separate bootstrap procedures: one for hyperparameter selection and another for dynamic identification. In both procedures, we compute $\hat{\tau }\_{tj}^{\circ}$ with bootstrap samples independently. For hyperparameter selection, $\hat{\tau }\_{tj}^{\circ}$, $\gamma$, and $\Delta$ are used to calculate $\hat{\tau }\_{tj}^*$ and select the hyperparameter pair $(c\_\texttt{L},c\_\texttt{R})$. For dynamic identification, $\hat{\tau }\_{tj}^{\circ}$ is used to identify the tie set in Eq (5), and this process does not depend on $\gamma$ and $\Delta$.
> > >
> > > - **Upper Bound of $\Delta$:** As previously mentioned, $\gamma$ is a small tuning parameter to ensure $\Delta < 1$. We have set the upper bound at 0.99 as a precautionary measure.
> > >
> > > - **Typo in New Results:** We apologize for the typo in Table 1. To address this issue, we are prepared to provide the code for the simulation to clarify any discrepancies.

---

### Official Review · Reviewer_pM3n · 2024-07-11

**Soundness:** 3
**Presentation:** 3
**Contribution:** 3
**Rating:** 7
**Confidence:** 4

**Summary:**

The paper introduces a dynamic subgroup identification strategy within the framework of covariate-adjusted response-adaptive randomization, addressing the need for more nuanced subgroup analysis in clinical trials. This strategy aims to optimize treatment allocation dynamically and identify subgroups that demonstrate significant treatment effects, which is crucial in practice.

**Strengths:**

1.	The method is highly relevant to modern clinical trial designs, where there is a pressing need to identify patient subgroups with differential treatment responses efficiently.
2.	The paper is strong in theoretical development, providing rigorous proofs and formulations that demonstrate the statistical validity and efficiency of the estimator for the best subgroup treatment effect.
3.	The approach looks novel, from my perspective.

**Weaknesses:**

1.	I understand the paper is a very technical paper, and the authors have spent a lot of efforts in making the paper easy to follow. My minor comment is that maybe providing more intuition on the reasons why the complicated design is needed will be helpful. For example, why we need resampling and bootstrap? What kind of technical challenge bootstrap can help to overcome?
2.	The objective function in line 128 seems a little mismatched with the objective of best subgroup identification. For example, if there are two arms in $\mathcal{T}_1$  with variance zero and very large respectively, it seems that optimizing the objective in line 128 will lead to the solution that we forget about the super large one and devote most efforts to the one with variance 0.  Thus, I am thinking about whether there is some analysis on the probability that we successfully identify $\mathcal{T}_1$? More specifically, can we always guarantee we identify everything in $\mathcal{T}_1$ and we never include something out of $\mathcal{T}_1$?

**Questions:**

See previous comments.

**Limitations:**

See previous comments.

---

> ### Author Rebuttal · Authors · 2024-08-06
>
> - Thank you for your encouragement. Below, we provide our understanding of why adaptive designs can be preferable in identifying the best subgroups compared to other alternatives, why resampling and bootstrapping are critical in our procedure, and what technical challenges bootstrapping addresses.
>
>   - To illustrate why adaptive designs can be preferable in identifying the best subgroups, we first review the literature on identifying subgroups and put our method into perspective. There are two lines of literature: The first line conducts a post-hoc analysis of already collected data from previous studies, while the second line focuses on adaptively collecting new data via randomized experiments to ascertain the best subgroup. Therefore, designing an adaptive data collection mechanism is the focus of the second line of literature, and our method aligns with the second line. We follow the second line for subgroup identification because While the first line of literature adopts post hoc subgroup analyses that do not require new data collection, they rely on untestable causal assumptions, limiting the credibility of causal conclusions (e.g., the unconfoundedness assumption is necessary for causal inference in observational studies, but unmeasured confounders can compromise these conclusions). Conversely, in randomized experiments, valid causal conclusions do not depend on such assumptions, but analysis of subgroup treatment effects may still face biases like the winner's curse, mainly if heterogeneous effects are selected from the data in an ad hoc manner (Guo and He, 2021;Andrews et al., 2019). Our approach aligns with the second line of literature and involves collecting data through randomized experiments to identify subgroups that benefit most from the treatment accurately. This approach is preferable because it avoids the limitations of analyzing existing data, which is the case with first-line approaches.  It ensures that treatments are randomly assigned, thus eliminating the influence of unmeasured confounders and allowing for robust, valid causal inferences without the need for untestable assumptions—distinctly superior to using observational data. Moreover, our adaptive design method permits iterative adjustments based on new information, enhancing trial flexibility and efficiency. We also hope to gently note that we are not deliberately constructing a complex adaptive experimental design strategy. Given randomized experiments can be time-consuming, costly, and potentially result in adverse patient outcomes if unsuccessful, our approach aims to efficiently allocate experimental resources within a limited budget to identify the most beneficial subgroup.
>
>   - Bootstrap and resampling-based methods play a crucial role when dealing with subgroups that have tied effect sizes in the population. To shed light on this issue, we consider a more straightforward scenario without adaptive data collection. In this simplified scenario, our objective is to identify the subgroup clusters (referred to as "tie sets" in our paper) and rank these clusters according to their average effect sizes with statistical confidence. Suppose we have in total $p$ subgroups forming $m$ clusters, and their population effect sizes and the averaged cluster effect sizes are $\underbrace{\beta\_1= \ldots =\beta\_k}\_{\text{Cluster 1}}, \beta\_{k+1}, \ldots, \beta\_{l-1},\underbrace{\beta\_{l} = \ldots = \beta\_{p}}\_{\text{Cluster m}}$, $\alpha\_{j} = \sum\_{l\in \text{Cluster j}} w\_l \beta\_l,  \sum\_{l\in \text{Cluster j}} w\_l = 1,  j = 1, \ldots, m$. Then, the bootstrap procedure proposed in our method helps us to achieve the following two goals simultaneously: Provide valid statistical inference (confidence intervals and consistent point estimates) on the ``ordered" averaged cluster effect sizes, that are $\alpha\_{(1)},\alpha\_{(2)}, \ldots, \alpha\_{(m)}$; Identify the clusters with a high probability. While numerous methods exist for subject clustering, providing confidence intervals for their ordered estimated mean effect sizes is a challenging task due to the winner's curse bias. This issue is well-known in the existing literature, where constructing confidence intervals for order statistics is particularly difficult. Therefore, our bootstrap procedure is important in identifying the tie set while delivering valid statistical inference on the identified best set of subgroups.
>
> - Thank you for your question. When there are two arms with the same treatment effects but different variances, our algorithm can effectively manage the allocation efforts, avoiding the scenario when most efforts are allocated to the arm with zero variance. We shall illustrate this in a two-stage adaptive experiment. At the start of Stage 1, we adopt the same treatment allocation for all subgroups. At the end of Stage 1, if there are two arms (or subgroups) in $\mathcal{T}\_1$  with similar treatment effects but one with zero variance and the other with large variance, our algorithm will be able to identify these two competing subgroups and then merge these two subgroups into the best "set"  $\hat{\mathcal{T}}\_1$. In Stage 2, since both of these two subgroups belong to $\hat{\mathcal{T}}\_1$, they will be assigned the same treatment allocation, denoted as $\hat{e}\_1^*$. Additionally, Theorem 1 in our manuscript provides an analysis of the probability of successfully identifying $\mathcal{T}\_1$. We show that as the sample size tends to infinity, we can always correctly identify the best set of subgroups. Therefore, we can guarantee the identification of everything in $\mathcal{T}\_1$ and ensure that nothing out of $\mathcal{T}\_1$ is included.
>
> Reference:
> - Andrews, I., Kitagawa, T., and McCloskey, A. (2019). Inference on winners. Technical report, National Bureau of Economic Research
> - Guo, X. and He, X. (2021). Inference on selected subgroups in clinical trials. Journal of the American Statistical Association, 116(535):1498–1506.

---

> > ### Comment · Reviewer_pM3n · 2024-08-08
> >
> > I really appreciate the authors efforts in clarifying my concerns, which are very helpful. Thanks!

---

### Author Rebuttal · Authors · 2024-08-06

We want to thank all of our reviewers for their very insightful suggestions and comments. We have made our best efforts to address the questions and comments raised by our reviewers.

To supplement our simulation studies and to provide additional information in response to some questions, we provide a pdf file which includes the following figures and a table:

- Figure 1: Comparison of the correct selection probability and normalized mutual information among three conventional methods and our proposed design strategy based on an additional dataset.
- Table 1: Simulation results that demonstrate the insensitivity of our method to the choice of tuning parameters.
- Figure 2: Comparison of the correct selection probability and normalized mutual information among causal tree model, complete randomization with AIPW estimator, contextual bandit algorithms including epsilon greedy algorithm and upper confidence bound 1 algorithm with AIPW estimator, and our proposed design with IPW estimator and AIPW estimator.

We greatly appreciate the reviewers for taking the time and effort to provide their valuable feedback.

---

### Comment · Area_Chair_kWEG · 2024-08-12
**Two days remain for discussion**

I thank you all for supporting NeurIPS 2024, and I extend my gratitude to the reviewers who have engaged in discussions with the authors. May I kindly remind the reviewers who have not yet responded to the authors’ rebuttals that two days remain to do so. Please review the authors' rebuttal and indicate whether it addresses your concerns and if you will be adjusting your scores.

AC

---

### Decision · Program_Chairs · 2024-09-25

**Decision:**

Accept (poster)

**Comment:**

This paper presents a design strategy to dynamically identify the best subgroups that demonstrate significant treatment effects in randomized trials. It theoretically demonstrates that the proposed design has a higher probability of identifying the best subgroups compared to conventional designs. The paper also validates the design using both synthetic and clinical trial data.

All reviewers agree that the proposed design is important in practice and novel. The design is supported by sound theoretical foundations. Reviewers have expressed some concerns about the presentation and explanations. Specifically, one reviewer raised concerns regarding the related work, presentation clarity, and the experimental comparison baselines. During the discussion phase, most of the reviewers' questions were addressed, and the remaining concerns do not affect the soundness or novelty of the paper. Therefore, the paper is acceptable.